# Bounding cross-shelf transport time and degradation in Siberian-Arctic land-ocean carbon transfer

Lisa Bröder [1,2,7], Tommaso Tesi [1,2,3], August Andersson[1,2], Igor Semiletov[4,5,6] & Örjan Gustafsson[1,2]

The burial of terrestrial organic carbon (terrOC) in marine sediments contributes to the regulation of atmospheric $CO_2$ on geological timescales and may mitigate positive feedback to present-day climate warming. However, the fate of terrOC in marine settings is debated, with uncertainties regarding its degradation during transport. Here, we employ compound-specific radiocarbon analyses of terrestrial biomarkers to determine cross-shelf transport times. For the World's largest marginal sea, the East Siberian Arctic shelf, transport requires $3600 \pm 300$ years for the 600 km from the Lena River to the Laptev Sea shelf edge. TerrOC was reduced by ~85% during transit resulting in a degradation rate constant of $2.4 \pm 0.6$ $kyr^{-1}$. Hence, terrOC degradation during cross-shelf transport constitutes a carbon source to the atmosphere over millennial time. For the contemporary carbon cycle on the other hand, slow terrOC degradation brings considerable attenuation of the decadal-centennial permafrost carbon-climate feedback caused by global warming.

[1] Department of Environmental Science and Analytical Chemistry, Stockholm University, 10691 Stockholm, Sweden. [2] Bolin Centre for Climate Research, Stockholm University, 10691 Stockholm, Sweden. [3] Institute of Marine Sciences—National Research Council, 40129 Bologna, Italy. [4] International Arctic Research Center, University Alaska Fairbanks, Fairbanks, AK 99775, USA. [5] Pacific Oceanological Institute, Russian Academy of Sciences, 690041 Vladivostok, Russia. [6] National Research Tomsk Polytechnical University, 634034 Tomsk, Russia. [7] Present address: Department of Earth Sciences, Vrije Universiteit Amsterdam, 1081 HV Amsterdam, Netherlands. Correspondence and requests for materials should be addressed to L.B. (email: l.m.broeder@vu.nl) or to Ö.G. (email: orjan.gustafsson@aces.su.se)

Anthropogenic perturbations have altered the global carbon cycle significantly[1,2]. One important component is the enhanced mobilization of soil organic carbon through land-use change, deforestation, and global warming[2]. In high-latitude regions, increasing soil permafrost thaw[3], accelerating coastal and sea floor erosion[4,5], and rising fluvial sediment discharge[6] are expected to amplify the delivery of terrestrial organic carbon (terrOC) to the Arctic Ocean. To what extent this dislocated material undergoes remineralization during transport and upon discharge, determines the intensity of this positive feedback mechanism to climate change, yet the fundamental processes are still insufficiently understood[2,7]. TerrOC sequestration in marine sediments transfers carbon from short-term reservoirs (atmosphere, oceans, and biosphere) to long-term storage (e.g., sedimentary rocks or petroleum)[8–11], a process which contributes to the regulation of atmospheric $CO_2$ levels. Continental shelves play a disproportionally important role as they account for ~80% of all OC burial, while making up less than 10% of the ocean area[2]. Nevertheless, there is an ongoing debate whether continental margins are net sinks or sources of carbon[1,12,13]. Mid-latitude to high-latitude shelves (>30° N and S) are generally considered to be carbon sinks, particularly with increasing $CO_2$ concentrations in the atmosphere[1,2,12]. However, bottom waters on the East Siberian Arctic Shelf, the World's largest shelf-sea system, are strongly oversaturated with $CO_2$, attributed to degradation of labile terrOC supplied from thawing permafrost[14].

Previous studies observed significant terrOC export from thawing permafrost to the Arctic shelves[15,16], with decreases in sedimentary bulk terrOC and terrestrial biomarkers with increasing distance from the coast[17–20]. These trends were mainly attributed to terrOC degradation during cross-shelf transport but, thus far, the timescale for such transport remains unconstrained, preventing rates and/or carbon fluxes to be derived from sedimentary terrOC degradation.

Here, we used the wide and shallow Laptev Sea shelf to investigate the fate of terrOC upon delivery from the Lena River. The Lena River in Northern Siberia is the 2nd largest freshwater source to the Arctic Ocean delivering the largest amounts of dissolved and particulate terrOC of a single Arctic river (~5.7 and ~0.8 Tg C per year, respectively), which corresponds to 18 and 14%, respectively, of the total delivery to the Arctic Ocean[21,22]. In particular, we aimed to constrain system-scale cross-shelf transport times and degradation rates for sedimentary terrOC. Sampling locations in this study span from the outlet of the Lena River, along a 600 km transect across the Laptev Sea, to the shelf edge (water depths from 4 to 92 m; Fig. 1). The East Siberian Arctic Shelf is the widest ocean margin on Earth and, thus, provides an extraordinary natural laboratory to constrain the age and fate of terrOC during its cross-shelf transport.

With this overarching goal in mind, we performed compound-specific radiocarbon analysis on terrestrial biomarkers (long-chain n-fatty acids with carbon chain-lengths 24, 26, 28 and 30, LCFA) sorbed on fine sediments (i.e., <63 μm). These lipids are derived from plant waxes and preferably bound to the fine fraction of the sediment, which is transported across the shelf[23]. Biomarkers generally comprise only a small portion of the total organic matter but maintain the radiocarbon signature of their source and may thus be employed to time various processes[24]. By dating molecules uniquely derived from terrestrial sources, here we circumvented age biases from (modern) marine organic matter and were able to determine the net cross-shelf transport time of sediment-associated terrOC.

## Results

### Cross-shelf transport times for sedimentary organic matter.
Sediment transport processes across continental margins are

generally described as hop-scotch scenarios[25], because the material is thought to undergo repeated cycles of burial and resuspension with potentially in situ ageing of several centuries before the next leap[13,25]. These leaps occur episodically, are often induced by storms, and are not unidirectional[13,25,26]. Other mechanisms for sediment transport in the Laptev Sea include the incorporation of suspended particulate material in sea ice during freeze-up, transport with the dense bottom water resulting from brine ejection, and with ocean currents. Depending on the prevailing atmospheric conditions of the Arctic Oscillation, Lena River waters are either largely transported parallel to the coastline towards the East Siberian Sea or across the shelf towards the Eurasian Basin of the Arctic Ocean[27]. The cross-shelf transport time discussed in this study should be understood as a net (unidirectional cross-shelf vector) transfer time and not the actual random-walk speed since the material very likely traveled a much longer total route than the net distance across the shelf.

Permafrost terrOC comprises a mixture of OC from late Pleistocene Ice Complex Deposits (ICD-PF, average age ~23 kyr, see also Methods) and from the (seasonally thawed) active layer (AL-PF, average age ~2.1 kyr, see also Methods). ICD-PF is mainly released by coastal erosion, while AL-PF is predominantly delivered by fluvial transport[15,28]. One earlier study on sediment collected close to the Lena River delta found a calibrated $^{14}C$ age of about 6300 years for the same terrestrial biomarkers used in this study (LCFA), suggesting substantial input of pre-aged permafrost terrOC[16]. This is confirmed by a calibrated $^{14}C$ age of 7100 years for the shallowest sample in this study (Table 1).

While bulk organic carbon ages become younger toward the outer shelf (from ~7.3 to ~3.5 kyr; explained by an increasing proportion of autochthonous marine organic matter with a modern radiocarbon age), terrestrial LCFA ages rise with increasing water depth (from ~6.3 to ~10.0 kyr, Fig. 2, Table 1). This trend is well described by a linear relationship:

$$\text{Age (kyr)} = a\left(\text{kyr m}^{-1}\right) \times \text{Water depth (m)} + b\ (\text{kyr}),$$
$$(R^2 = 0.75,\ p<0.05) \quad (1)$$

with fitted parameters $a = 0.039 \pm 0.004\ \text{kyr m}^{-1}$ and $b = 6.6 \pm 0.2$ kyr defining the inverse net cross-shelf transport velocity and the pre-ageing on land, respectively. Uncertainties were accounted for using Monte Carlo simulations (see Methods).

The increasing ratio of short-chain to long-chain FAs with increasing water depth indicates an increasing proportion of marine OC sources to the bulk TOC (Supplementary Fig. 1A), consistent with the trend towards higher (more enriched) stable carbon isotopic values ($\delta^{13}C$) for bulk TOC (Supplementary Fig. 2). The $\delta^{13}C$ values of LCFAs are constantly depleted (−31.2 ± 0.5‰), supporting the assumption of their terrestrial origin.

Stable hydrogen isotopes ($\delta^2H$) of long-chain n-alkanes may be used to differentiate between carbon fixed during glacial and interglacial periods (e.g., Tumara Paleosol Sequence in Northeast Siberia[29], as well as ICD-PF and AL-PF in Laptev Sea surface sediments[30]). The concentration-weighted average values of $\delta^2H$ for n-alkanes with carbon-chain lengths 27, 29, and 31 for the surface sediments along this Laptev Sea transect remain constant with water depth (Supplementary Fig. 3). We therefore infer that, along the transect, the terrOC sources, ICD-PF and AL-PF, stay fairly similar and that their relative contributions do not change considerably. The lack of significant trends in the mineralogical composition (Supplementary Table 1) and an extensive compilation of published data on clay mineral assemblages[31–33] of surface sediments support the assumption that there is no major change in sediment source along the transect (Supplementary Fig. 4).

Hydrodynamic sorting during sediment transport may affect the particle size distribution and thereby change the radiocarbon

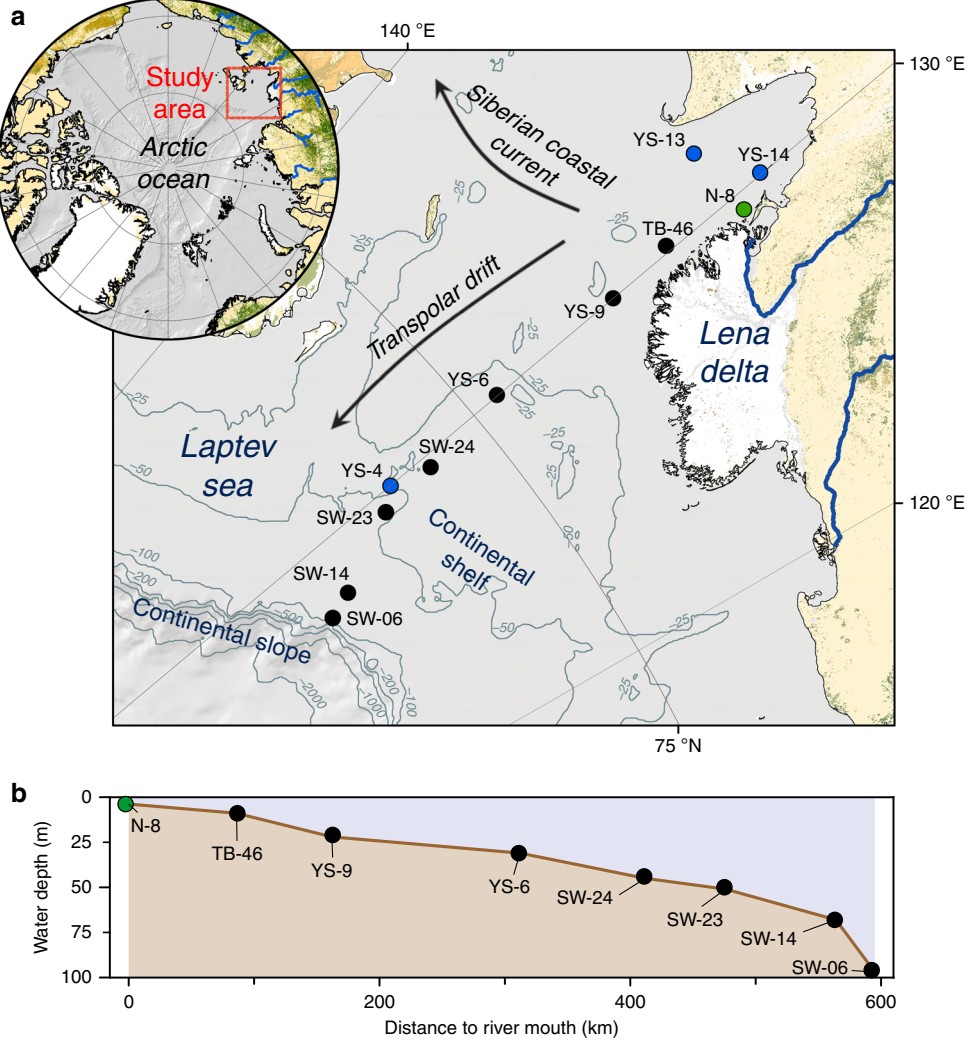

**Fig. 1** Map of the study area and depth-distance relationship of the sampling stations along the transect. Filled black circles refer to sampling locations on which radiocarbon dating of uniquely terrestrial molecules (biomarkers) was performed for this study; filled green circle: sampling station for radiocarbon data from a previous study[16], filled blue circles: sampling stations for additional biomarker data from an earlier study[19]. The underlying map was made with ArcGIS 10 using the latest version of the bathymetric grid IBCAO 3.0[59]. Arrows in (**a**) indicate net directions of the prevailing ocean currents. **b** The relationship between the distance from the river mouth (set as station N-8)

age if the different size fractions contain organic matter with different ages[23,34,35]. For sediments from the East Siberian Arctic Shelf and elsewhere, it has been shown that the lipid-rich OC associated with inorganic clastic particles (sorbed OC) is preferentially transported across the margin, while matrix free, coarse vascular plant fragments (rich in lignin) are deposited in shallow nearshore waters[23,36,37]. As a result, the relative distribution among size and density fractions changes during cross-shelf transport[23]. To circumvent the influence of hydrodynamic sorting our current study focused exclusively on the transport-prone fine fraction (particle sizes of <63 μm), carrying most of the OC load[23]. Within this fine fraction, the relative distribution of all biomarkers is relatively similar and the bulk OC radiocarbon age does not substantially differ between fine (settling velocity >1 m per day) and ultra-fine material (settling velocity <1 m per day) for the inner-shelf[23], implying that sorting within this fraction has a minimal effect. Additionally, the compound-specific hydrogen isotope measurements conducted on this size fraction suggest no noteworthy effect of hydrodynamic sorting. Therefore, the observed cross-shelf increase in LCFA ages can be attributed to terrOC ageing during lateral

transport as opposed to variations in source material or hydrodynamic sorting.

We defined the net cross-shelf transport time as

$$
\begin{aligned}
\text{Transport time (kyr)} &= \text{Age (kyr)} - b\,(\text{kyr}) \\
&= a\,(\text{kyr m}^{-1}) \times \text{Water depth (m)}
\end{aligned}
\tag{2}
$$

For the deepest and furthest offshore sampling station on the edge of the shelf break (station SW-06 at 92 m water depth, Fig. 1; Table 1), this resulted in a quantitative estimate of 3600 ± 300 years for the net transport time.

**Degradation of terrOC and biomarkers during transport.** Total terrOC was determined by source apportionment calculations with the stable and radiocarbon isotopic values for the bulk OC. This method makes use of the difference in carbon isotopic fingerprints for marine and terrestrial endmembers to assess their relative contributions to bulk OC (see Methods for more information). The approach has proven useful for sediments from this area in earlier studies[15,38]. There was an exponentially decreasing

**Table 1 Radiocarbon data for bulk organic carbon and long-chain fatty acids as well as the deduced transport times for stations along the Laptev Sea transect**

| Station ID | Water depth (m) | $\Delta^{14}C$ of TOC (‰) | ± (‰) | $^{14}C$ age TOC (kyr) | ± (kyr) | Calibrated CSRA age (kyr) | ± (kyr) | Transport time (kyr) | ± (kyr) |
|---|---|---|---|---|---|---|---|---|---|
| SW-06 | 92 | −364 | 2 | 3.57 | 0.02 | 9.99 | 0.34 | 3.56 | 0.32 |
| SW-14 | 64 | −314 | 2 | 2.97 | 0.02 | 8.63 | 0.42 | 2.48 | 0.23 |
| SW-23 | 56 | −333 | 2 | 3.19 | 0.02 | 8.47 | 0.16 | 2.17 | 0.18 |
| YS-4 | 50 | −437 | 3 | 4.56 | 0.09 | – | – | 1.93 | 0.17 |
| SW-24 | 46 | −284 | 2 | 2.62 | 0.02 | 9.67 | 0.23 | 1.78 | 0.16 |
| YS-6 | 32 | −465 | 3 | 4.97 | 0.09 | 8.86 | 0.15 | 1.23 | 0.11 |
| YS-9 | 23 | −415 | 6 | 4.25 | 0.09 | 7.33 | 0.07 | 0.89 | 0.08 |
| YS-13 | 19 | −543 | 2 | 6.23 | 0.09 | – | – | 0.74 | 0.07 |
| YS-14 | 7 | −504 | 2 | 5.58 | 0.09 | – | – | 0.27 | 0.03 |
| TB-46 | 6 | −436 | 2 | 4.54 | 0.06 | 7.10 | 0.19 | 0.23 | 0.02 |
| N-8 | 3.8 | – | – | – | – | 6.30 | 0.16 | – | – |

TOC denotes total, i.e., bulk, organic carbon and CSRA signifies compound-specific radiocarbon analysis, here performed on long-chain fatty acids. Data for station N-8 and $\Delta^{14}C$ of TOC have been published previously[16,19]

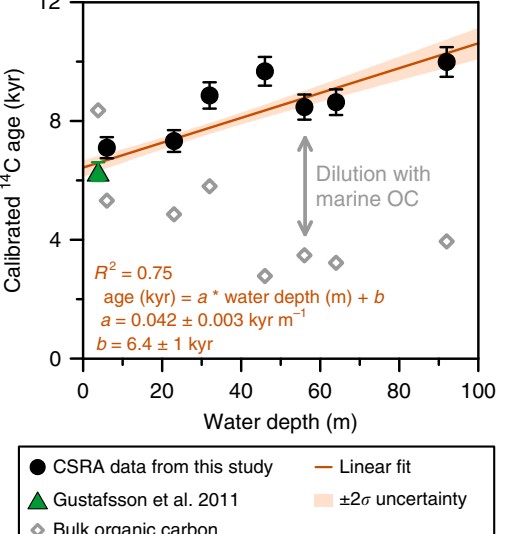

**Fig. 2** Calibrated radiocarbon ages of terrigenous long-chain *n*-fatty acids and bulk organic carbon vs. water depth. All filled symbols refer to compound-specific radiocarbon ages of long-chain *n*-fatty acids (LCFAs) with black circles: data from this study, green upward-pointing triangle: data point from a previous study[16], and error bars representing ± 1σ uncertainties. The linear fit to all biomarker ages (orange straight line, shaded area refers to ± 2σ uncertainties of the fit, determined by Monte Carlo simulations) is used to derive the transport time of sedimentary terrOC across the Laptev Sea shelf: Transport time (kyr) = Age (kyr) − *b* (kyr) = *a* (kyr m⁻¹) × Water depth (m). In contrast to the increasing ages with increasing water depth for the terrestrial biomarkers, bulk organic carbon ages (gray open diamonds) decrease due to a growing proportion of modern marine organic matter (see also Supplementary Fig. 1)

trend in surface-area normalized total terrOC loadings with increasing transport time (Fig. 3a). The same behavior was observed for the specific terrestrial biomarkers lignin phenols, cutin acids, long-chain *n*-fatty acids, and long-chain *n*-alkanes (Fig. 3c–f). A first-order degradation rate constant for each of the carbon pools was estimated from fitting a first-order kinetic reaction function to the data:

$$C(t) = C_{deg} \times e^{-kt} + R, \qquad (3)$$

where $C(t)$ is the concentration at time $t$, $C_{deg}$ is the initial concentration subtracted by the asymptotic offset, i.e., $C_{deg} = C(0) - R$, $k$ is the degradation rate constant and $R$ is the asymptotic value for $C(t \rightarrow \infty)$. Monte Carlo simulations, accounting for the combined uncertainties in terrOC and biomarker analyses and in transport times, yielded rates of 2.4 ± 0.6 kyr⁻¹ for terrOC and 2.8 ± 0.2, 2.6 ± 0.1, 4.0 ± 0.9, and 1.9 ± 0.4 kyr⁻¹ for the respective biomarkers lignin phenols, cutin acids, long-chain *n*-fatty acids, and long-chain *n*-alkanes (Fig. 3). The degradation rate constants of bulk terrOC and of these specific terrestrial biomarkers are thus fairly similar.

However, not all terrOC appears to degrade over this timescale of about 4000 years, as the exponential curves display an asymptotic positive offset ($C(t \rightarrow \infty) = R > 0$, gray-shaded area in Fig. 3a, c–f). The recalcitrant fraction $f_R$, defined as the offset divided by the initial value, i.e., $f_R = R/C(0)$, makes up 13 ± 4% for bulk terrOC (Fig. 3a). It has a similar proportion for long-chain *n*-alkanes (11 ± 3%), but is significantly smaller for lignin phenols, cutin acids, and long-chain *n*-fatty acids (2.6 ± 0.5, 0.9 ± 0.2, and 5 ± 2%, respectively; Fig. 3b–f).

## Discussion

Our study provides observation-based evidence that cross-shelf transport can be a millennial scale process. Specifically, we obtained a quantitative estimate of 3600 ± 300 years for the transport of sedimentary terrOC across the 600 km-wide Laptev Sea shelf upon supply from the Lena river.

The time spent during lateral transport, likely characterized by recurrent oxic-suboxic cycles driven by resuspension events, appears to be the regulatory parameter for sedimentary terrOC degradation as shown by the exponential decrease of bulk terrOC and terrestrial biomarker loadings with increasing transport times. This is consistent with and supports earlier hypotheses that call on the lateral oxygen exposure time (OET) to describe the cross-shelf gradient of reactive carbon pools[25,39]. However, previous estimates suffered from large uncertainties due to problems with constraining the transport times[25], which in this study has been tackled using compound specific radiocarbon dating[39].

Using this novel approach, the resulting first-order degradation rates of 2.4 ± 0.6 kyr⁻¹ for terrOC and 2.8 ± 0.2, 2.6 ± 0.1, 4.0 ± 0.9, and 1.9 ± 0.4 kyr⁻¹ for lignin phenols, cutin acids, long-chain *n*-fatty acids, and long-chain *n*-alkanes suggest rather slow degradation (Fig. 3). While these rates are all similar, the relative differences between the different carbon pools are consistent with previous findings. Specifically, earlier studies have suggested that terrOC in marine sediments is a complex mixture of multiple

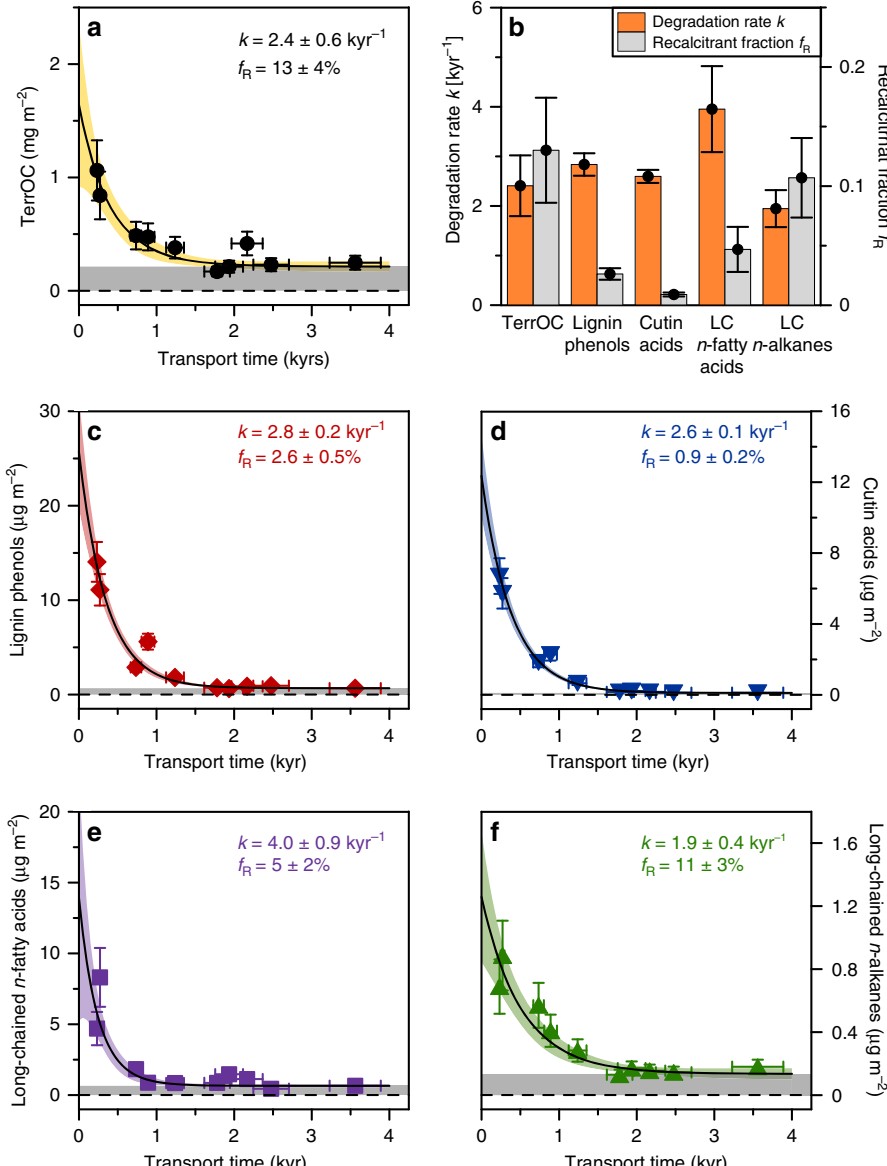

**Fig. 3** Degradation rates and recalcitrant fractions for different terrOC pools. An exponential decay curve $C(t) = C(0) \times e^{-kt} + R$ is fitted to measured loadings (filled symbols in **a**, **c**–**f**, error bars correspond to ±1σ uncertainties) in surface sediments vs. transport time (calculated from the age-depth relationship, Fig. 2). Displayed curves (black lines in panels **a**, **c**–**f**) and uncertainty ranges of the fit (shaded area correspond to ± 2σ in panels **a**, **c**–**f**) for all parameters were determined by Monte Carlo simulations. The gray areas in panels **a**, **c**–**f** mark the positive offset $C(t \to \infty) = R$, from which the recalcitrant fraction $f_R$ is calculated as $f_R = R/C(0)$. **a** TerrOC fraction of total organic carbon loadings as calculated with dual-carbon isotope source apportionment. **b** Comparison of the different first-order degradation rates $k$ and recalcitrant fractions $f_R$, derived in **a**, **c**–**f**. **c**–**f** Terrestrial biomarker loadings for the fine sediment fraction (<63 μm, see Methods section on bulk organic carbon and biomarker analyses and Supplementary Information Fig. 5 for details): lignin phenols (**c**), cutin acids (**d**), long-chain $n$-fatty acids (**e**), and long-chain $n$-alkanes (**f**)

substances with different reactivities, leading to a reactive continuum[26,40–42].

Furthermore, we decided to include a recalcitrant component (i.e., a fraction with a degradation rate constant of $0\,\mathrm{kyr^{-1}}$) to account for the limited changes observed for transport times >1.5 kyr. When computing degradation rate constants for the subset of inner-mid shelf data points where the transport time is shorter than 1.5 kyr, the rate constants are on average a factor of three higher than when using the full set of observations across the entire shelf (Supplementary Table 2). Results for the inner subset are thus still on the order of millennial scales, but have large uncertainties (standard deviations of 44–120%, compared to 4–25% when using the entire dataset; Supplementary Table 2).

Using this computational method of including an asymptotic positive offset (instead of computing several degradation rates for different timeframes/stretches of the shelf transect), our results show that different biomarkers displayed a varying fraction of the initial loadings to be refractory (characterized as a positive offset in our modeled kinetic reaction function). The fraction that did not degrade over a time span of 4000 years was $0.9 \pm 0.2\%$ for cutin acids, $2.6 \pm 0.5\%$ for lignin phenols, $5 \pm 2\%$ for long-chain $n$-fatty acids, and $11 \pm 3\%$ for long-chain $n$-alkanes. This corresponding fraction was $13 \pm 4\%$ for bulk terrOC, thus even higher than for all biomarkers except long-chain $n$-alkanes. It may be expected that substances with low degradation rates (i.e., highly recalcitrant, e.g., soot black carbon, kerogen) or intimate

associations with the mineral matrix are less readily extracted from the sediments and thus fall outside the analytical window for these specific biomarkers[43], but are included in our dual-isotope constrained bulk terrOC component. A recalcitrant terrOC fraction of 13 ± 4% (as constrained here) is consistent with earlier estimates for the neighboring East Siberian Sea[23], the East Siberian Arctic Shelf[44] and other shelf systems with expected long transport times (e.g., Amazon delta[45]). Taken together, this study also shows that degradation during cross-shelf transport occurs over millennial time scales.

An earlier attempt to quantify cross-shelf transport times on the Washington margin by tracing the volcanic ash of the 1980 Mount St. Helen eruption resulted in a transport time of <1 year[46]. This study may have underestimated the true terrOC transport time as this value stands in sharp contrast to another estimate for the same system that used bulk organic carbon $^{14}C$ measurements and assumptions on the proportion of terrOC in the bulk to derive a cross-shelf transport time of approximately 1800 years[25]. Furthermore, the use of compound-specific radiocarbon dating, as in our study on LCFAs in the mobile fraction (<63 μm) along the 600-km long Laptev Sea transect, circumvents uncertainties caused by e.g., changing proportions of marine organic matter and hydrodynamic sorting, and thus enables a quantitative constraint for terrOC transport across one of the widest ocean margins on Earth.

Altogether, our results demonstrate that the time spent during cross-shelf transport exerts first-order control on the fate of land-derived material in the marine environment. In sediment transport systems characterized by multiple settling-resuspension cycles, terrOC is repeatedly exposed to oxygen, which stimulates degradation[25]. For long distances such as across the Eurasian-Arctic shelves, the Amazon shelf, the New England shelf, and the North Sea shelf, lateral transport may last over centuries to millennia and would therefore be measurable on a radiocarbon timescale[13,45]. The quantified terrOC losses across continental margins result from slow degradation taking place over these long timescales. Hence, this functioning of the ocean margin carbon cycle attenuates the otherwise expected decadal-centennial scale carbon-climate feedback from vulnerable permafrost carbon pools.

At the same time, the current results document a potentially small, but persistent leakage of terrOC to short-term reservoirs on large continental margins, which stands in contrast to the rapid terrOC burial reported for e.g., tropical mountain rivers[47] or fjords around the globe[48]. With ongoing global warming causing permafrost thaw and hydrological changes, increasing delivery of terrOC is expected from both young/surface and old/deeper carbon pools[1,2,5,15,28]. For the Mackenzie River, the efficient transfer and burial of terrOC has been suggested as a geological $CO_2$ sink[9]. For the wide Eurasian Arctic shelves, on the other hand, long-lasting sediment transport allows for terrOC degradation, thereby constituting a carbon source to overlying water and atmosphere. Overall, our findings show that the effect of terrOC on atmospheric $CO_2$ concentrations over geological timescales may not be a simple function of the terrOC flux to the ocean, but depends largely on its further fate upon coastal delivery. In conclusion, sediment transport times across continental shelves are a key controlling factor determining whether mobilized terrOC becomes a source or a sink to the active carbon cycle.

## Methods

**Sampling**. The samples analyzed in this study have been collected during the Arctic expeditions ISSS-08 (The International Siberian Shelf Study) onboard the RV Yacob Smirnitskyi during summer 2008 and SWERUS C-3 (The Swedish-Russian-US Investigation of Carbon-Climate-Cryosphere Interactions in the East Siberian Arctic Ocean) on IB ODEN during summer 2014. Sediment cores were retrieved with an Oktopus multicorer (eight Plexiglas tubes, 10 cm diameter) or a GEMAX gravity corer (two Plexiglas tubes, 9 cm diameter) and surface sediments collected with a Van Veen grab sampler. For the grab samples only the uppermost cm was subsampled and used in this study. Sediment cores were cut into 1 cm slices within 24 h after sampling. All samples were kept frozen throughout the expedition and freeze-dried upon arrival to Stockholm University laboratories. For exact sampling locations see Supplementary Table 1.

**Bulk organic carbon and biomarker analyses**. Concentrations and stable carbon isotopes of total organic carbon (TOC) were analyzed at the Stable Isotope Laboratory in the Department of Geological Sciences, Stockholm University. Radiocarbon analysis of TOC was conducted at NOSAMS (National Ocean Sciences Accelerator Mass Spectrometry, Woods Hole Oceanographic Institution). Mineral surface area measurements and all biomarker analyses were performed at the Department of Environmental Science and Analytical Chemistry, Stockholm University. All bulk organic carbon, mineral surface area, and biomarker data also used in this study (compiled in Supplementary Table 3) have been previously used for other purposes and published; see Bröder et al.[19] for details on the methods.

The correction for hydrodynamic sorting during transport is new for this manuscript. Since the transport times were estimated for the fine fractions (<63 μm) and the present values were measured on the full size distribution (data from Bröder et al.[19]), the measured values of all biomarker and terrOC loadings were corrected for hydrodynamic sorting during transport by calculating the fraction fine in each sample with a relationship between water depth and fraction terrOC/biomarker in the fine sediments obtained using data from Tesi et al.[23], for details see Supplementary Fig. 5. Biomarkers and terrOC are normalized to the specific surface area, which is assumed to be a conservative parameter, to circumvent biases from other carbon pools[49,50].

**Compound-specific radiocarbon analysis**. Sediment samples from the mixed layer (top 3–4 cm, determined by XRF-measured Pb and Mn concentrations) were fractionated by particle size (wet sieving, cut-off 63 μm). Only the fine fraction was used for analysis to prevent age-biases from large particles that are retained close to the coast by hydrodynamic transport mechanisms[23].

Compound-specific radiocarbon dating was performed on fatty acid methylesters (FAMEs) with carbon chain-lengths 24, 26, 28 and 30 combined following the method by Eglinton et al.[51]. Lipid extraction was carried out using accelerated solvent extraction with DCM:MeOH (9:1). The extract was purified by activated copper and $Na_2SO_4$ (anh.) for sulfur and water removal, respectively, and separated into a neutral and an acid fraction using $NH_2$ Bond Elut columns. The acid fraction was then methylated with MeOH:HCl (95:5) at 70 °C for 12 h. After methylation, the hydrochloric acid was removed by wet extraction of the FAMEs (MilliQ water and n-hexane). Further clean-up, i.e., removal of unsaturated homologs, was done by column chromatography using $AgNO_3$-Si columns.

Long-chain FAMEs C24, C26, C28, and C30 were isolated using a preparative capillary gas chromatograph (pcGC) built around an Agilent 6890NGC system equipped with a DB-5 column (Agilent J&W, 60 m, 0.530 mm, film thickness 1.50 μm) and coupled with a Gerstel Cold Injection System (CIS3) and a preparative fraction collector (PFC)[51]. Compounds were separated using He as a carrier gas (7 ml min$^{-1}$) and a temperature program of 30–320 °C (temperature ramp of 20 °C min$^{-1}$ up to 130 °C, followed by 10 °C min$^{-1}$ up to 320 °C, hold time 22 min). The temperature of the transfer line and the PFC switch were kept at 320 °C through all injections. Approximately 1% of the column flow was diverted to a FID to monitor the separation and potential changes in retention times. The trapping windows were adjusted when necessary.

All targeted FAMEs were pooled in a single glass trap capillary and analyzed combined. The isolated compounds were rinsed from the trap with hexane and an aliquot (<2%) was analyzed by gas chromatograph-mass spectrometry (GC-MS) to evaluate the purity (between 94 and 99%, see Supplementary Table 4) and to assess if the isolated amounts were sufficient for radiocarbon analysis. An example chromatogram before and after pcGC isolation is shown in Supplementary Fig. 6. 20–50 consecutive injections (5 μl injection volume, FAMEs dissolved in toluene at concentrations of ~200 ng μl$^{-1}$) were needed to yield enough sample material (>20 μg C) for radiocarbon analysis, following standard routines at NOSAMS[52,53]. The pcGC program was performed in a clean (HEPA filtered room intake air) and over-pressurized laboratory dedicated to isotope work, and has previously been evaluated for low-level carbon influence on compound-specific radiocarbon results[54]. The radiocarbon ages reported by NOSAMS were corrected for the incorporation of radiocarbon-dead carbon atoms via methylation and then calibrated using the IntCal13[55] age model (assuming $\Delta R = 0$) to account for the temporal variability of the $\Delta^{14}C$ signature of atmospheric $CO_2$.

The proportions of the different homologs (C24, C26, C28, and C30) were rather constant, while the ratio of low-molecular-weight to high-molecular-weight FAMEs increased significantly across the shelf (Supplementary Fig. 1), pointing towards a negligible contribution of marine OC to the pooled FAMEs. Stable carbon isotope values of the long-chain FAMEs displayed no major trends with water depth and thus support the assumption of a terrestrial source for the isolated compounds (Supplementary Fig. 2).

Mollusc shells found in the surface and below the deepest sediment layer of the sediment cores used in this study (3–4 cm) were also sent for radiocarbon dating to constrain the depositional age (i.e., the time passed since sedimentation). All shells were found to have a modern $^{14}$C signature. Hence, we assumed the depositional ages to be negligible on a radiocarbon timescale (Supplementary Table 4).

**Compound-specific hydrogen isotope analysis**. All the ancillary compound-specific hydrogen isotope data used in this study have been published previously[30]. Compound-specific hydrogen isotope analysis was performed on long-chain n-alkanes (carbon chain-lengths 27, 29, and 31) by gas chromatography isotope-ratio monitoring mass spectrometry (GCirMS). For details on the method, we refer to Vonk et al.[30].

**Source apportionment**. The dual-carbon isotope signatures ($\Delta^{14}$C and $\delta^{13}$C) of bulk sedimentary OC were used to apportion the relative contributions from the three main sources: active layer permafrost (AL), ice complex deposit permafrost (ICD), and marine. The source-signatures (endmembers) were obtained from literature values: $\delta^{13}C_{AL} = -27.0 \pm 1.2$‰, $\delta^{13}C_{ICD} = -26.3 \pm 0.7$‰, $\delta^{13}C_{marine} = -21.0 \pm 2.5$‰, $\Delta^{14}C_{ICD} = -940 \pm 84$‰, and $\Delta^{14}C_{marine} = -50 \pm 12$‰[38]. The $\Delta^{14}C_{AL}$ endmember depends on the transport time, as $^{14}$C decays significantly on the timescale of a few thousand years, and this also reduces the endmember variability. Therefore transport-time dependent $\Delta^{14}C_{AL}$ values were used for each specific sample, calculated as follows: If $\tau$ is the lateral transport time in years, $\mu_0$ is the fraction modern for the AL endmember mean and $\sigma_0$ is the fraction modern standard deviation, then the dependence on $\tau$ is given by: $\mu_\tau = \mu_0 \times e^{-\tau/8033}$ and $\sigma_\tau = \sigma_0 \times e^{-\tau/8033}$. 8033 years is the mean decay time for $^{14}$C. The $\Delta^{14}$C-signature relates to the fraction modern ($f_m$) as: $\Delta^{14}C = \left(f_m \times e^{(1950 - yr)/8033} - 1\right) \times 1000$, where 1950 is the radiocarbon reference year and yr is the sampling year. The no-transport (near-shore) $\Delta^{14}C_{AL}$ value is $-232 \pm 147$‰ (average active layer depth: $36 \pm 20$ cm). However, after transport of $\tau = 3900$ years, this endmember becomes $-525 \pm 91$‰. The marine and ICD $\Delta^{14}$C endmembers were not corrected for transport since the marine contributions are from the near-present and because the ICD represents OC formed during the late Pleistocene and the aging for ICD $^{14}$C is therefore independent on if aged in the ocean or on land.

The AL and ICD fractions were combined to yield the total fraction terrestrial (fraction terrOC = fraction ICD + fraction AL = 1−fraction marine). To account for the variability of the endmember values, a Markov chain Monte Carlo (MCMC) approach was used, using in-house Matlab (ver. 2014b) scripts[56,57]. A total of 1 000 000 iterations were run for each sample, with a burn-in (initial search time) of 10 000 and a data thinning (removal of correlations between iterations) of 10. The variability between several MCMC runs is much smaller than the standard deviations of the estimated parameter distributions. Running the same simulation 10 times provides a relative standard deviation of less than 1% for the estimated parameter value. We therefore conclude that the convergence of the simulations does not affect our interpretations. The Monte Carlo simulations are very robust with respect to the choice of input parameters: regardless of their choice the runs converge within the burn-in phase. The stochastic perturbations are tuned to meet the well-established criterion of an acceptance rate of ~0.23, which theoretically has been found to be an optimal condition for a broad range of Markov chain Monte Carlo algorithms.[58]

**Calculation of transport time and degradation rate constants**. The lateral transport times of sedimentary terrOC across the Laptev Sea were estimated from calibrated compound-specific $^{14}$C ages of terrestrial biomarkers (long-chain n-fatty acids, LCFAs) for different sampling stations (Table 1). The transport time for a given sampling location was defined as the time of travel from the Lena River mouth (the main source of terrOC in the Laptev Sea) to the sampling station. However, a few sampling sites do not lie on the straight trajectory towards the north (e.g., south of the Lena River mouth) and it is therefore not clear exactly to which point the distance should be related. Instead, water depth is therefore used as a proxy for lateral transport distance. (Nevertheless, when using the distance from the sampling station closest to the Lena River mouth, we obtained a similarly good fit, Supplementary Fig. 7.) The fit was computed using a MCMC protocol, accounting for the age uncertainties from the age model.

The degradation rates for sedimentary terrOC and terrestrial biomarkers (lignin phenols, cutin acids, LCFAs, and long-chain n-alkanes) were estimated from the sampling site-specific transport times and the respective loadings. The data were fitted to an exponentially decaying function with a y-axis offset, see main text. The y-offset was required to obtain a good fit, and is explained by a recalcitrant fraction ($f_R$), i.e., a certain fraction that is non-degradable on this timescale. The fitting was done using a MCMC approach, accounting for the uncertainties of both the transport times (x-axis) and the concentrations of the investigated bulk terrOC and biomarker (y-axis).

**Data availability**. All data generated or analyzed during this study are included in this published article and its Supplementary Information. They will also be publicly available in Stockholm University's Bolin Center Database (https://bolin.su.se/data/Broder-2018).

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

## Acknowledgements

We thank crew and personnel of the IB ODEN, the RV Yakob Smirnitskyi and the TB0012. The Swedish-Russian-US Investigation of Carbon-Climate-Cryosphere Inter-actions in the East Siberian Arctic Ocean (SWERUS-C3) 2014 and the International Siberian Shelf Study 2008 (ISSS-08) expeditions were supported by the Knut and Alice Wallenberg Foundation, the Swedish Polar Research Secretariat and the Headquarters of the Far Eastern Branch of the Russian Academy of Sciences. This project was also supported by the Swedish Research Council (VR Contracts No. 621-2007-4631, 621-2013-5297 and 2017-01601), European Research Council (ERC-AdG CC-TOP project #695331 to Ö.G.), the US National Oceanic and Atmospheric Administration (OAR Climate Program Office, NA08OAR4600758/Siberian Shelf Study), the Russian Science Foundation (No. 15-17-20032), the Nordic Council of Ministers and the US National Science Foundation (OPP ARC 0909546; 1023281). L.B. also acknowledges financial support from the Climate Research School of the Bolin Climate Research Center. T.T. also acknowledges EU financial support as a Marie Curie fellow (contract no. PIEF-GA-2011-300259), contribution no. 1955 of ISMAR-CNR Sede di Bologna. I.S. additionally thanks the Russian Government for financial support (mega-grant #14.Z50.31.0012). This study was supported by the Delta Facility of the Faculty of Science, Stockholm University. Cecilia Bandh and Henry Holmstrand provided valuable assistance in the laboratory.

## Author contributions

L.B., T.T., I.S., and Ö.G. conceived of the approach and collected the samples; L.B. and T.T. performed the laboratory analyses; A.A. conducted Monte Carlo simulations; L.B., T.T., A.A., and Ö.G. interpreted the data; L.B. drafted the figures and all authors con-tributed to the final manuscript.

## Additional information

**Competing interests:** The authors declare no competing financial interests.

