## [Peer Review File · Nature Communications]

Reviewers' comments:

Reviewer #1 (Remarks to the Author):

This paper reports the timescale of cross shelf transport of terrestrial organic matter on the continental shelf off the Lena River mouth. They successfully conducted difficult tasks and I have positively read it. I think this paper is acceptable for the publication in Nature Communications. But I suggest following points to improve the manuscript.

1) Authors seem to assume that the ^{14}C age of a minor component (FAs) represents a major fraction of sediment. Please clearly explain the assumptions reside background of this study.

2) FA age presented in Fig. 2 should be a mean of those with a wide variety of ages. As an extreme case, FA age might simply reflect the weighted average of both modern and ^{14}C -dead FAs rather than the FAs with the age of Gaussian distribution around the average. Well, above hypothesis may be too extreme, but can the authors reject this hypothesis? This question is strongly related how to interpret the results.

3) How does bomb ^{14}C affect the FA age? If modern (<50 yr) FA contributes a lot in the sediment FA, the bomb ^{14}C significantly affect the result. The authors may roughly estimate it with their degradation rate.

4) I think $\delta^{14}\text{C}$ value makes sense rather than ^{14}C age as, for example, the vertical scale of Fig. 2. I strongly recommend to add the scale of $\delta^{14}\text{C}$ value as well as ^{14}C age.

5) How recent climate change affects the FA age in Fig. 2? As the authors mentioned, global warming dissolve permafrost during the last several decades, which should have added old (maybe ^{14}C dead) terrOC in the riverine transported OC.

6) The authors should mention about contamination from GC column bleed. I think open column chromatography is advisable after GC/PFC to remove column bleed.

Nao Ohkouchi

Reviewer #2 (Remarks to the Author):

This manuscript aims to quantitatively address the degradation and transport time of terrestrial organic matter deposited on continental margins in the Arctic Ocean. The main findings of this work suggest that ~85% of terrestrial organic carbon is lost along transport on the East Siberian shelf in the Arctic Ocean, with a transport time on a 1000-yr time scale. This has implications on the contribution that sedimentary terrestrial organic matter has for a net sink or source of terrestrial organic carbon within the active carbon cycle. This work aims at addressing a relevant and important paradigm (the “geochemical conundrum”) of understanding terrestrial OM losses from riverine discharge to final burial in ocean sediments (Hedges et al., 1997), in a system impacted by rapid changes in terrestrial organic matter translocation (ie. Through mobilization of soil organic carbon through permafrost thaw loss). This paper supports much of the previous work the authors have contributed the literature, and provides a direct estimate for terrestrial organic matter remineralization in the largest shelf sea of the Arctic Ocean. However, I feel that some of the description and details included in previous work (e.g. Bröder et al., 2016 Biogeochemistry) regarding methodology and/or assumptions taken for data analysis is lacking from the current manuscript. My main concerns, in regard to the validity of the current claims presented in this manuscript, are related to the calculation of bulk terrestrial organic carbon from the dual-carbon isotope analyses. This is not clearly touched upon at all in the main paper. Do you do a three-endmember mixing model to determine bulk terrestrial OC content? The explanation in the methods section (source apportionment) is a bit confusing; how can you correct for transport time of your radiocarbon signatures, when you are using the ^{14}C age to also determine this transport time? The use of the Monte Carlo simulations to model and optimize results seems to be a fine approach, but the authors should discuss any error analysis performed (i.e. how certain are you that you’ve properly optimized and parameterized your results?).

Overall, I feel that this paper will be of interest to the community and provides a novel contribution to the literature, which helps to quantitatively address the contribution of a possible ‘loss mechanism’ for terrestrial organic matter during along-shelf transport. I have outlined some clarification questions for the authors below. Particularly, I have outlined some details that should be included in the main body of the manuscript (some of these are addressed in the methods section, but I feel should be moved to the main discussion) to strengthen the claims the authors make in this paper.

Line 55- What is the proportion of carbon delivered to the Arctic Ocean from the Lena River (i.e. the Lena exports the most DOC ~5700 Gg yr⁻¹ (Holmes et al., 2012), and POC ~800 Gg yr⁻¹ (McClelland et al., 2016) of a single arctic river)? See eg. Holmes et al., 2012 (DOC) and McClelland et al., 2016 (POC) for Pan-Arctic and river specific estimates.

This gives a bit of context to how much carbon is delivered to this system, and its importance on a larger basin-scale setting (i.e. Arctic Ocean).

Lines 61-67- While your justifications for using long-chained n-fatty acids are valid, a few questions immediately came to my mind when reading the main text: 1) How do you ensure no marine sources to the fatty acids (i.e. chain lengths used, isotopic composition, etc.)? 2) Why is it better to use these

biomarkers rather than compound specific work on other “terrestrially-unique” biomarkers (ie. Lignin phenols, cutin acids, tannins, etc.)? Each set of biomarkers certainly have their own caveats, but the choice of the fatty acids needs to be completely convincing when first addressed.

Line 79- What is the total particle transport along this transect? How does this compare to other shelf regions in the Arctic and around the world? How might this affect your conclusions and their applications in other systems? (i.e. importance of deposition vs. lateral transport for net carbon sequestration)

Lines 83-90- Do you have end-member values for CSRA on long-chained n-fatty acids? How does CSRA for fatty acids of end-members compare to the bulk ages of permafrost OM and active layer SOM?

Lines 99-108- What is the likely source based on all of these results? (also, extra “and” in parenthetical statement on line 101- should be “...Siberia, ICD-PF and AL-PF”)

Line 117- Reference for the fine fraction carrying the majority of the OC load, especially in this region? How might the discussion on lines 109-122 affect the overall interpretations of your results, i.e. the degradation of sedimentary terrOC? In other words: Would you expect the mineral-bound OC to be more or less reactive than the matrix free materials? Does this strengthen your conclusions of sedimentary terrOC degradation importance as a C source to the atmosphere? Since you’ve studied a fraction not impacted by hydrodynamic sorting, what scenario does your study represent (i.e. upper bound of sedimentary terrOC as a C source to the atmosphere, lower bound, etc)?

Line 128-131- How do you determine total terrOC? How do you account for taxa-related differences in production and contribution of different biomarker you measure to terrestrial OC? ie. contribution of n-alkanes to sediments is very different dependent on taxa (e.g. angiosperms vs gymnosperms (Lane, 2017 Organic Geochemistry)).

Line 168- Again, I believe how you determined the contribution of bulk terrestrial OC warrants discussion within the text of the paper!

Lines 173-183 This discussion is a bit wordy and the argument isn’t clear. Are you just setting precedence, or are you arguing that the transport time is greater on one of the largest margins in any ocean (this makes sense...)?

Line 187 “stimulated” should be “stimulates”?

Lines 191-193 How does this tie into the proportion of OM transported from the terrestrial to aquatic in this region? For example, if most of the organic matter (including permafrost OM) is transported as DOM, how does this impact this idea of carbon-climate feedbacks in the system?

Line 201 I would argue that this should be “an ultimate” CO₂ source to the atmosphere. How do you know that this degradation is complete to CO₂? And how (where, when) does the supersaturation of CO₂ at depth outgas to the atmosphere?

Materials and Methods

The flow of this section (i.e ordering of subsections) seems a bit jumbled. For example, bulk organic carbon mineral surface area and biomarker analyses section could be after sampling, but before the compound-specific work; and the compound specific sections should be sequential.

Line 219- In this compound-specific radiocarbon analysis section, you cite the methods for the dating protocol before the extraction. This would flow better to move sentence from lines 220-221 down to the start of the paragraph at line 226.

Lines 297-301 While the description of usage of the Monte Carlo simulations for the transport timing, degradation and recalcitrant fraction fits was clear to me, I am not sure I understand how the Monte Carlo simulations were used for the source appropriation. Could the authors comment a bit more on their input variables, assumptions and the error associated with this methodology?

References:

Bröder, L., Tesi, T., Salvadó, J.,A., Semiletov, I. P., Dudarev, O. V., & Gustafsson, Ö. (2016). Fate of terrigenous organic matter across the laptev sea from the mouth of the lena river to the deep sea of the arctic interior. *Biogeosciences*, 13(17), 5003-5019. doi:<http://dx.doi.org/10.5194/bg-13-5003-2016>

Hedges, J. I., Keil, R. G., & Benner, R. (1997). What happens to terrestrial organic matter in the ocean? *Organic geochemistry*, 27(5), 195-212.

Holmes, R. M., McClelland, J. W., Peterson, B. J., Tank, S. E., Bulygina, E., Eglinton, T. I., ... & Staples, R. (2012). Seasonal and annual fluxes of nutrients and organic matter from large rivers to the Arctic Ocean and surrounding seas. *Estuaries and Coasts*, 35(2), 369-382.

Lane, C. S. (2017). Modern n-alkane abundances and isotopic composition of vegetation in a gymnosperm-dominated ecosystem of the southeastern US coastal plain. *Organic Geochemistry*, 105, 33-36.

McClelland, J. W., Holmes, R. M., Peterson, B. J., Raymond, P. A., Striegl, R. G., Zhulidov, A. V., ... & Staples, R. (2016). Particulate organic carbon and nitrogen export from major Arctic rivers. *Global Biogeochemical Cycles*, 30(5), 629-643.

Reviewer #3 (Remarks to the Author):

Manuscript Title: Bounding cross-shelf transport time and degradation 1 in Siberian-Arctic land ocean carbon transfer

Authors: Lisa Bröder, Tommaso Tesi, August Andersson, Igor Semiletov, Örjan Gustafsson

This paper describes the state-of-the-art application of compound specific radiocarbon analyses of n-alkanes to constrain the transport and degradation time for terrestrial OC (terrOC) on the Laptev Sea shelf. The authors analyze their data using a clever approach that provides important insights about the fate of terrOC in a rapidly changing system (Arctic Shelf). My main concern about the work is that the rate constants (not rates; see below) are fit to data for 0 to 4 kyr timeframes. However, in most cases, the loading data for TerrOC and biomarkers remain constant between ~1.5 to 4 kyr, which may artificially make the rate constants more conservative. Thus, I recommend that the authors recompute the rate constants for a 0 to 1.5 kyr timeframe and compare these rate constants with those computed over the 4 kyr timeframe. Since organic matter decomposition changes over time (see Middelburg (1989) *Geochimica et Cosmochimica Acta* Vol. 53, pp. 1577-1581), averaging over 4 kyr likely underestimates the rate constants and transit times.

I would also like to see the authors to acknowledge that the rate constants presented in their manuscript are averaged over a kyr timeframe. In reality, the highest rates of decomposition occur soon after deposition and averaging over long timescales likely underestimates the reactivity of terrOC. Another important point is that degradation of terrOC such as lignin only occurs when oxygen is present so oxygen exposure time (OET), a parameter that may be embedded in transport time, is an important determinant of the role of terrOC in the carbon cycle.

Overall, this is a unique and interesting data set that provides new insights about the transport time and fate of terrOC across the Siberian-Arctic shelf. I recommend publication after considering the issue mentioned above as well as some rewriting of the manuscript. In some places, the manuscript is worded awkwardly and the word choices could be more direct. I've provided detailed comments below that the authors should consider when revising their manuscript.

Editorial Comments:

Abstract should be rewritten. Choice of wording is awkward and the findings should be presented more directly and succinctly.

Line 21-22 and Line 41. Change to “that regulates atmospheric CO₂” or “contributes to the regulation of atmospheric CO₂”

Line 24-25. Revise to, “compound specific radiocarbon analyses of terrestrial biomarkers to date cross-shelf transport times”

Line 27-28. Revise to, “TerrOC was reduced by 85% during transit resulting in a degradation rate constant of $2.4 \pm 0.6 \text{ kyr}^{-1}$ ”

Line 28. What do the authors mean by “protracted transport”?

Line 42. Omit “only”.

Line 52. Omit “left”. Change to “remains unconstrained”.

Line 62 and throughout text including figure captions

. The correct language is “long-chain n-fatty acids, LCFA”, not “long-chained”. The authors may want to note that long-chain fatty acids derive from plant waxes, but I suggest using an alternative to “wax lipids”, which could be confused with “wax esters”.

Line 66. Use alternative wording for “clock”, which is not customarily used as a verb (e.g., “date”, or “determine the net transport time”)

Line 73. Authors should define what they mean by “potentially significant in situ aging”. This language is vague and should be omitted or revised.

Line 74. Revise to, “and are not unidirectional”.

Lines 102-103. This paragraph opens by discussing hydrogen isotopes as source tracers but the authors discuss their findings as “concentrated weighted average n-alkanes”. Instead, I recommend that the authors describe their findings as “concentrated weighted average values of $\delta^{2}\text{H}$ for C27, C29 and C31 n-alkanes”. Revise description of findings to be more direct (e.g., “remained constant with water depth” rather than “does not display a significant trend with water depth”).

Line 114. Revise to “are deposited in shallow waters” rather than “trapped”.

Line 119. Define “fine” vs. “ultra-fine”.

Line 121. Avoid using “significant” here and throughout text unless statistical support is provided.

Line 133. The authors present “first-order rate constants” (units of per time) rather than “rates” (units of changes in concentration per unit time). The text and equation terms should be revised accordingly.

Line 137. I think this should read terrOC and biomarker analyses. The present wording suggests the ratio of terrOC to biomarkers (terrOC/biomarkers).

Line 149-150. Specify that cross-shelf transport FOR THIS SYSTEM is a millennial scale process. Without additional information, it is unclear whether rates for the Laptev Sea shelf can be extended to other systems.

Figure 1. Revise y-axis label to read, “ $\delta^{2}\text{H}$ of long-chain n-alkanes [‰]. “HMW” is not defined and is unnecessary because the n-alkanes are already defined as “long-chain”.

Figure 3. Cite Fig. S4 as support for statement, “due to a growing proportion of modern marine organic matter.”

Author responses to reviewer comments, and edits to manuscript number NCOMMS-17-20661 "Bounding cross-shelf transport time and degradation in Siberian-Arctic land-ocean carbon transfer" by L. Bröder, T. Tesi, A. Andersson, I. Semiletov, and Ö. Gustafsson

Reviewer #1 (Nao Ohkouchi)

GENERAL COMMENTS:

"This paper reports the timescale of cross shelf transport of terrestrial organic matter on the continental shelf off the Lena River mouth. They successfully conducted difficult tasks and I have positively read it. I think this paper is acceptable for the publication in Nature Communications."

We appreciate and are encouraged by the positive assessment and have paid attention to the constructive suggestions. Please find below our detailed responses (and actions in response) to each of the raised issues.

SPECIFIC POINTS:

- 1) *"Authors seem to assume that the ^{14}C age of a minor component (FAs) represents a major fraction of sediment. Please clearly explain the assumptions reside background of this study."*

We chose these biomarkers because previous work had shown that FAs (as well as other lipids) are mainly associated with the mineral-bound fine fraction that is preferentially transported (e.g., Tesi et al., 2016). Also, long-chain FAs are common biomarkers for terrOC and the method for compound-specific radiocarbon analysis on these compounds is well established (e.g., Eglinton et al., 1996; Mollenhauer and Rethemeyer, 2009).

Importantly, we are not assuming that the ^{14}C ages of these terrestrial biomarkers are representing the ^{14}C ages of the bulk OC. Instead, we actually show in Fig. 2 that the opposite is the case: while the ^{14}C ages of the FAMES increase with increasing water depth, the bulk OC ^{14}C ages decrease due to a growing proportion of marine OC. This difference illustrates the power in the source-specificity provided by molecular- ^{14}C analysis. Furthermore, we acknowledge that biomarkers in general only make up a small portion of the total OC (Line 65-67).

- 2) *"FA age presented in Fig. 2 should be a mean of those with a wide variety of ages. As an extreme case, FA age might simply reflect the weighted average of both modern and ^{14}C -dead FAs rather than the FAs with the age of Gaussian distribution around the average. Well, above hypothesis may be too extreme, but can the authors reject this hypothesis? This question is strongly related how to interpret the results."*

This is a good clarifying point; any CSRA measurement is representing the average for a population of the target molecule, which naturally has a distribution of ^{14}C ages. Hence, one cannot entirely rule out changing proportions of young vs old FAs along the transect. However, the associated lack of changes in their $\delta^{13}\text{C}$ signature (Supplementary Fig. 2), the

relative contributions of the different long chain-lengths (Supplementary Fig. 1B) and in the δD of the long-chain alkanes (Supplementary Fig. 3) all provide pieces of evidence/information, supporting a fairly constant proportion along the transect. Taken all these data together, it is therefore reasonable to hypothesize that the trend towards higher FA ages with increasing water depth is indeed caused by ageing of the entire population during transport rather than changing proportions (as two identical molecules of different ^{14}C ages still have the same physical and chemical properties yielding similar processing).

- 3) *“How does bomb ^{14}C affect the FA age? If modern (<50 yr) FA contributes a lot in the sediment FA, the bomb ^{14}C significantly affect the result. The authors may roughly estimate it with their degradation rate.”*

The relatively high age of the FAs already close to the shore of >6000 years suggests a negligible influence of bomb ^{14}C . Generally, for this region terrestrial carbon sources are thought to have long residence times which is reflected in their mean radiocarbon ages for TOC (ice-complex deposits ~23 kyr and active layer ~2.1 kyr, see also Materials and Methods). Marine OC sources are affected by bomb ^{14}C , but they do not contribute to the long-chain FAs and thus do not affect the LCFA ages.

- 4) *“I think [capital delta] ^{14}C value makes sense rather than ^{14}C age as, for example, the vertical scale of Fig. 2. I strongly recommend to add the scale of [capital delta] ^{14}C value as well as ^{14}C age.”*

We agree that $\Delta^{14}C$ can provide useful extra information, however, to measure the cross-shelf transport time, years (or in this case kyr) seemed the more appropriate unit. Since we are using calibrated ^{14}C ages there is no linear conversion to $\Delta^{14}C$; therefore, we could not simply add a $\Delta^{14}C$ scale to Fig. 2. However, we did consider this comment and have thus moved this central radiocarbon data from the Supplementary Information to the main text. The new Table 1 now provides the radiocarbon ages of the FAs, as well as the $\Delta^{14}C$ values for the bulk OC in addition to the TOC radiocarbon ages.

- 4) *“How recent climate change affects the FA age in Fig. 2? As the authors mentioned, global warming dissolve permafrost during the last several decades, which should have added old (maybe ^{14}C dead) terrOC in the riverine transported OC.”*

One expects that, with ongoing warming and continued permafrost thaw, increasing amounts of pre-aged terrOC will be released and eventually end up in shelf sediments. With transport times on the order of millennia, recent climate change events (on the order of decades) have likely not yet progressed far across the shelf. It is conceivable that the station close to land could have received a higher proportion of old permafrost material in recent times. However, the stable hydrogen isotope values for long-chain *n*-alkanes and the mineralogical composition suggest that any such changing input is at most of minimal influence to the make-up of the current system (Supplementary Fig. 3 and 4, Supplementary Table 1).

- 5) *“The authors should mention about contamination from GC column bleed. I think open column chromatography is advisable after GC/PFC to remove column bleed.”*

This is a good point. Open column chromatography as a mean to remove potential contamination from column bleed has been tested on some of these samples. However, small sample sizes, low recoveries and comparably little effect on the purity of the isolated compounds ultimately prevented us from using it. We carefully monitored the purity of the isolated compounds by running the samples on the GC-FID prior to AMS analyses, the purities were fortunately high, and these results are reported in Supplementary Table 2.

Reviewer #2

GENERAL COMMENTS:

“This manuscript aims to quantitatively address the degradation and transport time of terrestrial organic matter deposited on continental margins in the Arctic Ocean. The main findings of this work suggest that ~85% of terrestrial organic carbon is lost along transport on the East Siberian shelf in the Arctic Ocean, with a transport time on a 1000-yr time scale. This has implications on the contribution that sedimentary terrestrial organic matter has for a net sink or source of terrestrial organic carbon within the active carbon cycle. This work aims at addressing a relevant and important paradigm (the “geochemical conundrum”) of understanding terrestrial OM losses from riverine discharge to final burial in ocean sediments (Hedges et al., 1997), in a system impacted by rapid changes in terrestrial organic matter translocation (ie. Through mobilization of soil organic carbon through permafrost thaw loss). This paper supports much of the previous work the authors have contributed the literature, and provides a direct estimate for terrestrial organic matter remineralization in the largest shelf sea of the Arctic Ocean. However, I feel that some of the description and details included in previous work (e.g. Bröder et al., 2016 Biogeochemistry) regarding methodology and/or assumptions taken for data analysis is lacking from the current manuscript. My main concerns, in regard to the validity of the current claims presented in this manuscript, are related to the calculation of bulk terrestrial organic carbon from the dual-carbon isotope analyses. This is not clearly touched upon at all in the main paper. Do you do a three-endmember mixing model to determine bulk terrestrial OC content? The explanation in the methods section (source apportionment) is a bit confusing; how can you correct for transport time of your radiocarbon signatures, when you are using the ^{14}C age to also determine this transport time? The use of the Monte Carlo simulations to model and optimize results seems to be a fine approach, but the authors should discuss any error analysis performed (i.e. how certain are you that you’ve properly optimized and parameterized your results?).

Overall, I feel that this paper will be of interest to the community and provides a novel contribution to the literature, which helps to quantitatively address the contribution of a possible ‘loss mechanism’ for terrestrial organic matter during along-shelf transport. I have outlined some clarification questions for the authors below. Particularly, I have outlined some details that should be included in the main body of the manuscript (some of these are addressed in the methods section, but I feel should be moved to the main discussion) to strengthen the claims the authors make in this paper.”

We are pleased about the overall positive evaluation and would like to thank the reviewer for the clear, concise and constructive comments. We agree with the majority of the comments and suggestions for further clarification and have revised the manuscript accordingly, as outlined in the responses to the more detailed comments below.

Regarding the first specific comment here on the source apportionment, we have now included an expanded explanation in the main text (Line 138-142). In brief, we are working with two different sets of radiocarbon data: The radiocarbon ages of the terrestrial biomarkers (FAs) are used to determine the cross-shelf transport time, while the $\Delta^{14}\text{C}$ values, along with the $\delta^{13}\text{C}$, of the total organic carbon (TOC) are used in the source apportionment calculations, following the well-established method applied also in earlier studies (e.g., Bröder et al., 2016; Tesi et al., 2016a; Vonk et al., 2012).

The second general comment concerned the statistical source apportionment method. The Markov chain Monte Carlo approach used in several earlier and ongoing studies is detailed also in the earlier papers, yet perhaps best in the paper by Andersson et al. (2015). It builds on the observation that to obtain statistically sound source estimates from mass balance relations the endmember distributions and data uncertainties need to be taken into account, not only for uncertainty estimation but also for correct estimates of the central values (Andersson, 2011). For the present case, the absolutely largest uncertainty comes from the numerical spread of the endmember distributions. With the current setup, using 1 000 000 iterations, a burn-in of 10 000 iterations and a data thinning of 10, the variability of the runs is much smaller than the standard deviations of the estimated parameter distributions. Running the same simulation 10 times provides a relative standard deviation of less than 1% for the estimated parameter value. Thus, the convergence of the simulations does not affect our interpretations. These factors have been well examined in our previous publications (e.g., Andersson et al., 2015; Fang et al., 2017). More details on the Monte Carlo strategy have now also been included in the Materials and Methods section (Line 335-342).

SPECIFIC POINTS:

- 1) *“Line 55- What is the proportion of carbon delivered to the Arctic Ocean from the Lena River (i.e. the Lena exports the most DOC ~5700 Gg yr⁻¹ (Holmes et al., 2012), and POC ~800 Gg yr⁻¹ (McClelland et al., 2016) of a single arctic river)? See eg. Holmes et al., 2012 (DOC) and McClelland et al., 2016 (POC) for Pan-Arctic and river specific estimates. This gives a bit of context to how much carbon is delivered to this system, and its importance on a larger basin-scale setting (i.e. Arctic Ocean).”*

We agree that these facts provide useful background information to the reader and have now inserted the following sentence (Line 53-56): “The Lena River in Northern Siberia is the 2nd largest freshwater source to the Arctic Ocean delivering the largest amounts of dissolved and particulate terrOC of a single Arctic river (~5.7 and ~0.8 Tg C per year, respectively), which corresponds to 18 % and 14 %, respectively, of the total delivery to the Arctic Ocean (Holmes et al., 2012; McClelland et al., 2016).”

- 2) *“Lines 61-67- While your justifications for using long-chained n-fatty acids are valid, a few questions immediately came to my mind when reading the main text: 1) How do you ensure no marine sources to the fatty acids (i.e. chain lengths used, isotopic composition, etc.)? 2) Why is it better to use these biomarkers rather than compound specific work on other “terrestrially-unique” biomarkers (ie. Lignin phenols, cutin acids, tannins, etc.)? Each set of biomarkers certainly have their own caveats, but the choice of the fatty acids needs to be completely convincing when first addressed.”*

We understand that the answers to the reviewer’s questions were mainly found in the Materials and Methods section and not in the main manuscript. We have therefore lifted some information from the Materials and Methods section to the main text (Line 62-65 and Line 101-105) and hope that this clarifies matters more efficiently.

Regarding the first point, the exclusivity of the long-chained FA biomarkers: The FA chain-lengths isolated in this study for radiocarbon measurements were 24, 26, 28 and 30 carbon atoms, commonly employed terrestrial biomarkers; FAs produced by marine plankton have lower chain-lengths. As can be seen in Supplementary Fig. 1A, the ratio between short- and long-chain FAs clearly changes with increasing water depth, due to a growing proportion of marine OC. The proportions of the long-chain FAs, on the other hand, do not display such a trend (Supplementary Fig. 1B). The same pattern is observed for $\delta^{13}\text{C}$ values (Supplementary Fig. 2). Marine OC is generally more enriched, explaining the trend towards higher $\delta^{13}\text{C}$ values for bulk OC, while $\delta^{13}\text{C}$ values for the FAs are constantly low throughout the transect.

Regarding the second question about other terrestrial biomarkers: For our study area a recent publication, Tesi et al., 2016, showed that long-chain FAs are mainly associated with (i.e., bound to) the fine sediment fraction that is transported across the shelf, while a large fraction of the lignin phenols is found in matrix-free plant fragments that are retained close to the coast due to the large diameters of these fragments. For this reason, even though both lignin phenols and FAs are terrestrial biomarkers, we focused on the mobile fraction which is better represented by FAs to measure the cross-shelf transport time. Also, compound-specific radiocarbon analysis on FAs is a well-established method that had been tested on several samples from this region in earlier studies (Gustafsson et al., 2011; Vonk et al., 2014), yet with a different scope/application than in this present study.

- 3) *“Line 79- What is the total particle transport along this transect? How does this compare to other shelf regions in the Arctic and around the world? How might this affect your conclusions and their applications in other systems? (i.e. importance of deposition vs. lateral transport for net carbon sequestration)”*

All of these are valid and good questions. The only numbers on total particle transport we could find for the study area stem from box models by Stein and Fahl (in Stein and Macdonald, 2004). They estimate transport across the Laptev Shelf via sea ice, bottom currents, brines, etc. to add up to about 52 Pg yr^{-1} , which corresponds to about 60 % of the annual input. Sedimentation rates obtained from ^{210}Pb measurements similarly suggest a fairly high deposition also on the outer shelf (Salvadó et al., 2017; Vonk et al., 2012). As we

state in the main text, this study was conducted on one of the Earth's widest continental margins and the transport times for narrower shelves are likely to be shorter. For instance, for the Beaufort Sea, another Arctic shelf, rapid burial has been reported (Hilton et al., 2015) due to the different geomorphology of the margin. There, in contrast to the Laptev Sea, shelf sediments are viewed as a carbon sink. The separate question on the relative importance of deposition compared to degradation during lateral transport for the Laptev Sea was beyond the scope of the present study but is in focus for ongoing and future studies.

- 4) *“Lines 83-90- Do you have end-member values for CSRA on long-chained n-fatty acids? How does CSRA for fatty acids of end-members compare to the bulk ages of permafrost OM and active layer SOM?”*

We do not have such data and could not find any published values either. It must be stressed, though, that end-member CSRA data is not necessary for the current application of the CSRA data to “time” cross-shelf transport.

- 5) *“Lines 99-108- What is the likely source based on all of these results? (also, extra “and” in parenthetic statement on line 101- should be “...Siberia, ICD-PF and AL-PF)”*

We have now changed that part to read: “We therefore infer that along the transect the terrOC sources, ICD-PF and AL-PF, stay fairly similar and that their relative contributions do not change considerably.” (Line 11-112) Here, the extra “and” was meant to show that these are two different studies. For clarification, we have now changed that parenthetic statement to read: “Tumara Paleosol Sequence in Northeast Siberia²⁶, as well as ICD-PF and AL-PF in Laptev Sea surface sediments²⁷”.

- 6) *“Line 117- Reference for the fine fraction carrying the majority of the OC load, especially in this region?”*

The missing reference (Tesi et al., 2016b) has been inserted here.

- 7) *“How might the discussion on lines 109-122 affect the overall interpretations of your results, i.e. the degradation of sedimentary terrOC? In other words: Would you expect the mineral-bound OC to be more or less reactive than the matrix free materials? Does this strength your conclusions of sedimentary terrOC degradation importance as a C source to the atmosphere? Since you’ve studied a fraction not impacted by hydrodynamic sorting, what scenario does your study represent (i.e. upper bound of sedimentary terrOC as a C source to the atmosphere, lower bound, etc)?”*

The novel approach here of quantitatively constraining cross-shelf transport times of terrestrial tracers opens up the possibility to quantitatively deduce the relative importance of exposure to oxygen during lateral transport versus oxygen exposure times after burial. To this end, we focused on the mineral-bound phase that is largely transported across the shelf as opposed to the matrix-free coarse material that is predominantly deposited closer to the shore. One could expect this matrix-free fraction to be more reactive as it lacks mineral-phase associations that are thought to stabilize OC (e.g., Lalonde et al., 2012). In an earlier study,

however, we did not see substantial degradation after burial on a centennial scale (Bröder et al., 2016). We therefore infer that the time spent during transport plays a crucial role for OC degradation. Hydrodynamic sorting may also affect OC degradation patterns, but it seems difficult to assess in what way without any further data. Hence, we would prefer to refrain from categorizing this study as an upper or lower boundary in terms of C release to the atmosphere.

- 8) *“Line 128-131- How do you determine total terrOC? How do you account for taxa-related differences in production and contribution of different biomarker you measure to terrestrial OC? ie. contribution of n-alkanes to sediments is very different dependent on taxa (e.g. angiosperms vs gymnosperms (Lane, 2017 Organic Geochemistry)).”*

Total terrOC was determined by means of source apportionment calculations using bulk carbon isotope values $\delta^{13}\text{C}$ and $\Delta^{14}\text{C}$ as in several earlier studies (e.g., Tesi et al., 2016a; Vonk et al., 2012). Biomarker data have not been used to calculate terrOC and thus taxa-related differences in biomarker production are therefore not/less relevant to our approach. We understand that more information on the source apportionment calculations should be included in the main text and not only in the Materials and Methods section. We have now added a short paragraph (Line 138-142).

- 9) *“Line 168- Again, I believe how you determined the contribution of bulk terrestrial OC warrants discussion within the text of the paper!”*

We acknowledge an apparent lack of clarity with regards to the source apportionment calculations and have now added and clarified information in the main text (Line 138-142). Please see also our previous and later comments on this topic.

- 10) *“Lines 173-183 This discussion is a bit wordy and the argument isn’t clear. Are you just setting precedence, or are you arguing that the transport time is greater on one of the largest margins in any ocean (this makes sense...)?”*

This paragraph was meant to compare our study to previous attempts for constraining cross-shelf transport times. We have now shortened it substantially for clarity.

- 11) *“Line 187 “stimulated” should be “stimulates?”*

Correct, thank you for pointing this out. It has now been changed.

- 12) *“Lines 191-193 How does this tie into the proportion of OM transported from the terrestrial to aquatic in this region? For example, if most of the organic matter (including permafrost OM) is transported as DOM, how does this impact this idea of carbon-climate feedbacks in the system?”*

This is an interesting question, yet beyond the scope of the current paper, which has a focus on the transport and degradation in the sedimentary compartment during cross-shelf transport. We are currently working on estimates for a total “degradation flux” of terrOC in

both the sedimentary and water column (DOC and suspended POC) systems on the East Siberian Arctic Shelf.

- 13) *“Line 201 I would argue that this should be “an ultimate” CO₂ source to the atmosphere. How do you know that this degradation is complete to CO₂? And how (where, when) does the supersaturation of CO₂ at depth outgas to the atmosphere?”*

This is a valid point. We have changed that part since this study is not specifically investigating if the degraded sedimentary terrOC is completely turned into CO₂ and then outgassed. The sentence now reads as follows: “For the wide Eurasian Arctic shelves, on the other hand, long-lasting sediment transport allows for terrOC degradation, thereby constituting a carbon source to overlying water and atmosphere.”

- 14) *“Materials and Methods: The flow of this section (i.e ordering of subsections) seems a bit jumbled. For example, bulk organic carbon mineral surface area and biomarker analyses section could be after sampling, but before the compound-specific work; and the compound specific sections should be sequential.”*

The order of the different subsections has been changed according to these suggestions.

- 15) *“Line 219- In this compound-specific radiocarbon analysis section, you cite the methods for the dating protocol before the extraction. This would flow better to move sentence from lines 220-221 down to the start of the paragraph at line 226.”*

We have now moved this sentence accordingly.

- 16) *“Lines 297-301 While the description of usage of the Monte Carlo simulations for the transport timing, degradation and recalcitrant fraction fits was clear to me, I am not sure I understand how the Monte Carlo simulations were used for the source appropriation. Could the authors comment a bit more on their input variables, assumptions and the error associated with this methodology?”*

The major uncertainties regarding the Monte Carlo based source apportionment calculations are associated with the underlying assumptions: that isotopic mass balance is fulfilled, that the endmembers values properly reflect the sources, that there is no isotopic fractionation etc. The Monte Carlo simulations in themselves are very robust with respect to the choice of input parameters: regardless of choice the runs converge within the burn-in phase. The stochastic perturbations are tuned to meet the well-established criterion of an acceptance rate of ~ 0.23, which theoretically has been found to be an optimal condition for a broad range of Markov chain Monte Carlo algorithms (Roberts et al., 1997). These factors were established in previous work (e.g., Andersson et al., 2015; Keskitalo et al., 2017). Please see also our response to the General Comments. We have included more details on the Monte Carlo strategy both in the main text (Line 138-142) and in the Materials and Methods section (Line 335-342).

Reviewer #3

GENERAL COMMENTS:

“This paper describes the state-of-the-art application of compound specific radiocarbon analyses of n-alkanes to constrain the transport and degradation time for terrestrial OC (terrOC) on the Laptev Sea shelf. The authors analyze their data using a clever approach that provides important insights about the fate of terrOC in a rapidly changing system (Arctic Shelf). My main concern about the work is that the rate constants (not rates; see below) are fit to data for 0 to 4 kyr timeframes. However, in most cases, the loading data for TerrOC and biomarkers remain constant between ~1.5 to 4 kyr, which may artificially make the rate constants more conservative. Thus, I recommend that the authors recompute the rate constants for a 0 to 1.5 kyr timeframe and compare these rate constants with those computed over the 4 kyr timeframe. Since organic matter decomposition changes over time (see Middelburg (1989) Geochimica et Cosmochimica Acta Vol. 53, pp. 1577-1581), averaging over 4 kyr likely underestimates the rate constants and transit times.

I would also like to see the authors to acknowledge that the rate constants presented in their manuscript are averaged over a kyr timeframe. In reality, the highest rates of decomposition occur soon after deposition and averaging over long timescales likely underestimates the reactivity of terrOC. Another important point is that degradation of terrOC such as lignin only occurs when oxygen is present so oxygen exposure time (OET), a parameter that may be embedded in transport time, is an important determinant of the role of terrOC in the carbon cycle.

Overall, this is a unique and interesting data set that provides new insights about the transport time and fate of terrOC across the Siberian-Arctic shelf. I recommend publication after considering the issue mentioned above as well as some rewriting of the manuscript. In some places, the manuscript is worded awkwardly and the word choices could be more direct. I’ve provided detailed comments below that the authors should consider when revising their manuscript.”

We are naturally delighted about the positive appraisal and appreciate the useful comments and suggestions. We fully agree with the notion of non-constant degradation rates and thus the need to be clearer on the fact that, in this study, the apparent first-order degradation rate constants refer to an average over a kyr scale. We have also followed the reviewer’s suggestion of computing the degradation rate constants for data points where the transport time is <1.5 kyr, and compared those with our previous estimates using all data points:

	TerrOC (kyr ⁻¹)	Lignin phenols (kyr ⁻¹)	Cutin acids (kyr ⁻¹)	LC n-alkanes (kyr ⁻¹)	LC n-fatty acids (kyr ⁻¹)
Data points <1.5 kyr	8.1±5.0	9.3±4.0	3.7±2.4	8.5±3.7	10±12
All data points	2.4±0.6	2.8±0.2	2.6±0.1	1.9±0.4	4.0±0.9

On a first sight, it seems the reviewer's hypothesis is correct: using the <1.5 kyr data alone provides higher rates, on average about a factor of three higher than when using all data points. However, the estimated standard deviations are also much larger, on average 15 times larger than the standard deviations estimated for the full data set. This is not surprising: In total, we have 10 data points, of which five are <1.5 kyr, which are fitted to an offset exponential function with three parameters. Thus, the <1.5 kyr case (with five data points) does not constrain the decay rate well enough for us to be able to draw the conclusion that we indeed observe a change in degradation rate constant over time. A much larger dataset would be needed to properly test this hypothesis. In order to avoid oversimplification on the other hand, we had introduced the "recalcitrant fraction" as a parameter quantifying the portion that remains, which is largely governed by the data points with ages >1.5 kyr. Overall, even if degradation rates were higher by a factor of three for the first millennia, this would not change the conclusions of our study that terrOC degradation on this wide Arctic shelf takes place on a timescale of centuries to millennia. Nevertheless, since we agree with the notion, we have added a paragraph on this topic to the main text (Line 180-189) and the table shown above to the Supplementary Information (Supplementary Table 2).

We agree that the oxygen exposure time (OET) plays a major role in terms of terrOC degradation. This is particularly relevant for carbon pools that are predominantly reactive in oxic conditions like e.g., lignin phenols. As explained in the text, the terrestrial material that reaches the outer Laptev Sea shelf experienced millennial-scale transport which translates into a prolonged oxygen exposure time. On the one hand, we cannot really assess the actual time spent in oxic conditions because we cannot entirely exclude temporary burial below the top oxic layer of the sediments before the next mobilization event. One could therefore argue that our transport time should be considered an upper OET limit. On the other hand, it is reasonable to assume that this fraction escaped burial to be transported all the way to the outer shelf, thus, the OET in first-order approximation likely scales with the transport time. More on the OET concept has now been included in the manuscript (Line 166-173).

We have followed all of the editorial comments and our respective responses can be found below.

SPECIFIC POINTS:

- 1) *"Abstract should be rewritten. Choice of wording is awkward and the findings should be presented more directly and succinctly."*

We have revised the abstract according to the following points made by the reviewer. Please see our responses to the specific points 2-5.

- 2) *"Line 21-22 and Line 41. Change to "that regulates atmospheric CO₂" or "contributes to the regulation of atmospheric CO₂""*

These sentences have been changed accordingly.

- 3) *“Line 24-25. Revise to, “compound specific radiocarbon analyses of terrestrial biomarkers to date cross-shelf transport times””*

This has been changed to “compound specific radiocarbon analyses of terrestrial biomarkers to determine cross-shelf transport times”.

- 4) *“Line 27-28. Revise to, “TerrOC was reduced by 85% during transit resulting in a degradation rate constant of $2.4 \pm 0.6 \text{ kyr}^{-1}$ ””*

This sentence has been revised.

- 5) *“Line 28. What do the authors mean by “protracted transport”?”*

The word “protracted” has been changed to “long-lasting” for clarification.

- 6) *“Line 42. Omit “only”.”*

This word has now been removed.

- 7) *“Line 52. Omit “left”. Change to “remains unconstrained”.”*

This sentence has been changed accordingly.

- 8) *“Line 62 and throughout text including figure captions. The correct language is “long-chain n-fatty acids, LCFA”, not “long-chained”. The authors may want to note that long-chain fatty acids derive from plant waxes, but I suggest using an alternative to “wax lipids”, which could be confused with “wax esters”.”*

Thank you for pointing this out. “Long-chained” has been changed to “long-chain” throughout the text, figure captions and Supplementary Information. Instead of “wax lipids” it now reads “These lipids are derived from plant waxes and preferably bound to the fine fraction of the sediment”. This hopefully clarifies matters.

- 9) *“Line 66. Use alternative wording for “clock”, which is not customarily used as a verb (e.g., “date”, or “determine the net transport time”)”*

We have now changed “clock” into “determine”.

- 10) *“Line 73. Authors should define what they mean by “potentially significant in situ aging”. This language is vague and should be omitted or revised.”*

This part has been revised and now reads as follows: “the material is thought to undergo repeated cycles of burial and resuspension with potentially in situ ageing of several centuries before the next leap”.

- 11) *“Line 74. Revise to, “and are not unidirectional”.”*

The word “are” has been added here.

- 12) *“Lines 102-103. This paragraph opens by discussing hydrogen isotopes as source tracers but the authors discuss their findings as “concentrated weighted average n-alkanes”. Instead, I recommend that the authors describe their findings as “concentrated weighted average values of $\delta^2\text{H}$ for C27, C29 and C31 n-alkanes”. Revise description of findings to be more direct (e.g., “remained constant with water depth” rather than “does not display a significant trend with water depth”).”*

This part has been changed according to the suggested edits. It now reads: “The concentration-weighted average values of $\delta^2\text{H}$ for n-alkanes with carbon-chain lengths 27, 29 and 31 for the surface sediments along this Laptev Sea transect remain constant with water depth (Supplementary Fig. 3).”

- 13) *“Line 114. Revise to “are deposited in shallow waters” rather than “trapped”.”*

The word “trapped” has been changed to “deposited”.

- 14) *“Line 119. Define “fine” vs. “ultra-fine”.”*

The differentiation between “fine” and “ultra-fine” particles was done by settling velocity with a cutoff at 1 m per day using SPLITT fractionation (Tesi et al., 2016b). This information has now been added (Line 127-128).

- 15) *“Line 121. Avoid using “significant” here and throughout text unless statistical support is provided.”*

The word “significant” has been replaced here (Line 121) and also in lines 27, 75, 178, and 301.

- 16) *“Line 133. The authors present “first-order rate constants” (units of per time) rather than “rates” (units of changes in concentration per unit time). The text and equation terms should be revised accordingly.”*

This is correct, we have made the according changes throughout the manuscript.

- 17) *“Line 137. I think this should read terrOC and biomarker analyses. The present wording suggests the ratio of terrOC to biomarkers (terrOC/biomarkers).”*

Thank you for pointing this out. It now says “terrOC and biomarker analyses”.

- 18) *“Line 149-150. Specify that cross-shelf transport FOR THIS SYSTEM is a millennial scale process. Without additional information, it is unclear whether rates for the Laptev Sea shelf can be extended to other systems.”*

We agree that this first sentence of the discussion was ambiguous. We have now revised it to read: “Our study provides observation-based evidence that cross-shelf transport can be a millennial scale process.”

- 19) *“Figure 1. Revise y-axis label to read, “ $\delta^{2}H$ of long-chain n-alkanes [‰]. “HMW” is not defined and is unnecessary because the n-alkanes are already defined as “long-chain”.”*

Supplementary Fig. 3 (which was previously Fig. S1) and its caption have been revised accordingly.

- 20) *“Figure 3. Cite Fig. S4 as support for statement, “due to a growing proportion of modern marine organic matter.””*

Thank you for this good suggestion. A reference to Supplementary Fig. 1 (which was previously Fig. S4) has now been added to the caption of Fig. 2 (which we assume was meant here instead of Fig. 3).

REFERENCES

- Andersson, A.: A systematic examination of a random sampling strategy for source apportionment calculations, *Sci. Total Environ.*, 412–413, 232–238, doi:10.1016/j.scitotenv.2011.10.031, 2011.
- Andersson, A., Deng, J., Du, K., Zheng, M., Yan, C., Sköld, M. and Gustafsson, Ö.: Regionally-Varying Combustion Sources of the January 2013 Severe Haze Events over Eastern China, *Environ. Sci. Technol.*, 49(4), 2038–2043, doi:10.1021/es503855e, 2015.
- Bröder, L., Tesi, T., Andersson, A., Eglinton, T. I., Semiletov, I. P., Dudarev, O. V., Roos, P. and Gustafsson, Ö.: Historical records of organic matter supply and degradation status in the East Siberian Sea, *Org. Geochem.*, 91, 16–30, doi:10.1016/j.orggeochem.2015.10.008, 2016.
- Eglinton, T. I., Aluwihare, L. I., Bauer, J. E., Druffel, E. R. M. and McNichol, A. P.: Gas chromatographic isolation of individual compounds from complex matrices for radiocarbon dating, *Anal. Chem.*, 68(5), 904–912, doi:10.1021/ac9508513, 1996.
- Fang, W., Andersson, A., Zheng, M., Lee, M., Holmstrand, H., Kim, S.-W., Du, K. and Gustafsson, Ö.: Divergent Evolution of Carbonaceous Aerosols during Dispersal of East Asian Haze, *Sci. Rep.*, 7(1), 10422, doi:10.1038/s41598-017-10766-4, 2017.
- Gustafsson, Ö., Van Dongen, B. E., Vonk, J. E., Dudarev, O. V. and Semiletov, I. P.: Widespread release of old carbon across the Siberian Arctic echoed by its large rivers, *Biogeosciences*, 8(6), 1737–1743, doi:10.5194/bg-8-1737-2011, 2011.
- Hilton, R. G., Galy, V., Gaillardet, J., Dellinger, M., Bryant, C., O'Regan, M., Gröcke, D. R., Coxall, H., Bouchez, J. and Calmels, D.: Erosion of organic carbon in the Arctic as a geological carbon dioxide sink, *Nature*, 524(7563), 84–87, doi:10.1038/nature14653, 2015.
- Holmes, R. M., McClelland, J. W., Peterson, B. J., Tank, S. E., Bulygina, E., Eglinton, T. I., Gordeev, V. V., Gurtovaya, T. Y., Raymond, P. a., Repeta, D. J., Staples, R., Striegl, R. G., Zhulidov, A. V. and Zimov, S. a.: Seasonal and Annual Fluxes of Nutrients and Organic Matter from Large Rivers to the Arctic Ocean and Surrounding Seas, *Estuaries and Coasts*, 35(2), 369–382, doi:10.1007/s12237-011-9386-6, 2012.
- Keskitalo, K., Tesi, T., Bröder, L., Andersson, A., Pearce, C., Sköld, M., Semiletov, I. P., Dudarev, O. V. and Gustafsson, Ö.: Sources and characteristics of terrestrial carbon in Holocene-scale sediments of the East Siberian Sea, *Clim. Past*, 13(9), 1213–1226, doi:10.5194/cp-13-1213-2017, 2017.
- Lalonde, K., Mucci, A., Ouellet, A. and Gélinas, Y.: Preservation of organic matter in sediments promoted by iron, *Nature*, 483(7388), 198–200, doi:10.1038/nature10855, 2012.
- McClelland, J. W., Holmes, R. M., Peterson, B. J., Raymond, P. A., Striegl, R. G., Zhulidov, A. V., Zimov, S. A., Zimov, N., Tank, S. E., Spencer, R. G. M., Staples, R., Gurtovaya, T. Y. and Griffin, C. G.: Particulate organic carbon and nitrogen export from major Arctic rivers, *Global Biogeochem. Cycles*, 30(5), 629–643, doi:10.1002/2015GB005351, 2016.

- Mollenhauer, G. and Rethemeyer, J.: Compound-specific radiocarbon analysis – Analytical challenges and applications, *IOP Conf. Ser. Earth Environ. Sci.*, 5, 12006, doi:10.1088/1755-1307/5/1/012006, 2009.
- Roberts, G. O., Gelman, A. and Gilks, W. R.: Weak convergence and optimal scaling of random walk Metropolis algorithms, *Ann. Appl. Probab.*, 7(1), 110–120, 1997.
- Salvadó, J. A., Bröder, L., Andersson, A., Semiletov, I. P. and Gustafsson, Ö.: Release of Black Carbon from Thawing Permafrost Estimated by Sequestration Fluxes in the East Siberian Arctic Shelf Recipient, *Global Biogeochem. Cycles*, doi:10.1002/2017GB005693, 2017.
- Stein, R. and Fahl, K.: The Laptev Sea: Distribution, Sources, Variability and Burial of Organic Carbon, in *The Organic Carbon Cycle in the Arctic Ocean*, edited by R. Stein and R. W. Macdonald, pp. 213–236., 2004.
- Stein, R. and Macdonald, R. W., Eds.: *The organic carbon cycle in the Arctic Ocean*, Springer Verlag., 2004.
- Tesi, T., Muschitiello, F., Smittenberg, R. H., Jakobsson, M., Vonk, J. E., Hill, P., Andersson, A., Kirchner, N., Noormets, R., Dudarev, O., Semiletov, I. and Gustafsson, Ö.: Massive remobilization of permafrost carbon during post-glacial warming, *Nat. Commun.*, 7, 13653, doi:10.1038/ncomms13653, 2016a.
- Tesi, T., Semiletov, I., Dudarev, O., Andersson, A. and Gustafsson, Ö.: Matrix association effects on hydrodynamic sorting and degradation of terrestrial organic matter during cross-shelf transport in the Laptev and East Siberian shelf seas, *J. Geophys. Res. Biogeosciences*, 121(3), 731–752, doi:10.1002/2015JG003067, 2016b.
- Vonk, J. E., Sánchez-García, L., van Dongen, B. E., Alling, V., Kosmach, D., Charkin, A., Semiletov, I. P., Dudarev, O. V., Shakhova, N., Roos, P., Eglinton, T. I., Andersson, A. and Gustafsson, Ö.: Activation of old carbon by erosion of coastal and subsea permafrost in Arctic Siberia, *Nature*, 489(7414), 137–140, doi:10.1038/nature11392, 2012.
- Vonk, J. E., Semiletov, I. P., Dudarev, O. V., Eglinton, T. I., Andersson, A., Shakhova, N., Charkin, A., Heim, B. and Gustafsson, Ö.: Preferential burial of permafrost-derived organic carbon in Siberian-Arctic shelf waters, *J. Geophys. Res. Ocean.*, 119(12), 8410–8421, doi:10.1002/2014JC010261, 2014.

REVIEWERS' COMMENTS:

Reviewer #1 (Remarks to the Author):

I think the authors properly replied at least my comments, and filled the gap of understanding of the reported data between they and me. The manuscript was also improved, and is now acceptable in Nature Communications.

Reviewer #2 (Remarks to the Author):

In general, I believe that the authors adequately addressed the points raised by the 3 reviewers in this revised manuscript. I find that the revised paper reads much clearer in the current state, especially with regards to the approaches taken for data analysis. I have a few additional, yet minor, syntax suggestions on the current manuscript that may further help with clarity before publication.

Line 21- I feel like the sentence starting with "Here, we leverage..." is missing the object of what you are specifically leveraging, i.e. the utilization of a method to address a gap in knowledge. Perhaps would read better as "Here, we leverage the utility of compound-specific..."

Line 24- Perhaps would be better as "TerrOC, as determined through isotopic mixing model, was..."

Line 25-26- "long-lasting" is a bit vague. Perhaps better as "cross-shelf" and/or "net"?

Line 34- do you need to redefine "terrOC" here, as it has only been defined in the Abstract?

Line 41-42 I would say that this also depends on the type of shelf! Might be useful at some point in introduction or conclusions to tie in the idea that this type of work can be applied to other systems to better understand variability in these cross-shelf processes on terrOC "sequestration" Also, see comment on lines 210-216 below in regards to this suggestion.

Line 45 "attributed to" instead of "due"

Line 48 comma after the citations "15,16" and "in" instead of "of"

Line 52- "the fate of terrOC" instead of "terrOC fate"

Line 67- Omit "here"

Line 68- add "here" before "we"

Line 104- Maybe give an average and standard deviation of the ^{13}C values of the LCFA here?

Line 106- I would recommend removing "Isotopic ratios of...", and start the sentence with "Stable hydrogen isotopes..."

Line 129- "Additionally" instead of "Also"

Line 131- Add "Therefore" to start the sentence "The observed cross-shelf..."

Line 132- add something along the lines of "as opposed to variations in source or hydrodynamic sorting" to the end of this sentence

Line 136- Add "net transport time" after "quantitative estimate"

Lines 157-158 and lines 182-184 seemingly contradict each other. Clarify what is meant in each statement to resolve this apparent contradiction.

Lines 187-195- These lines address one of the main concerns of reviewer #3, however, I feel the explanation in the text is still a bit confusing to follow. Perhaps can alter the order this is presented from the analytical progression you took to its overall implications - i.e. I understood this section as follows: you included the recalcitrant component to deal with the idea of multiple reactivates across the shelf, perhaps due to OET, mineral protection, etc., as evident by the 3X rate of degradation that might be occurring within the first 1.5kyr that the material is transported, compared to the net transport time of 3.6kyr of terrOC transport across the shelf.

How exactly do the three sentences from lines 201-216 tie together? I would perhaps address this as how estimates of cross shelf transport times benefit from these types of analyses e.g. "Previous attempts to quantify cross-shelf transport times in Washington margin using volcanic ash from the 1980 Mount St. Helen eruption (ref 46) may underestimate true terrOC cross-shelf transport times comparatively to bulk OC radiocarbon techniques (ref 25). Furthermore, the utilization of compound-specific biomarker analyses presented in this study allows for [pick up on line 216]..." This might also be better incorporated if included in the introduction instead, as motivation for why a better approach is needed for quantitatively estimating cross shelf transport times.

Line 228- Add "on large continental margins"

Line 235- The effect of what? Add "terrOC has..." between "effect" and "on"

Line 237- start the sentence "Sediment..." with a "Therefore," Or "Thus," Or "In conclusion,"

Figure 3. I would just reiterate in this figure caption (especially C-F) that the biomarker data was corrected for what was 'present' in the fine fraction, which is outlined in Figure S5.

Reviewer #3 (Remarks to the Author):

This paper describes the state-of-the-art application of compound specific radiocarbon analyses of n-alkanes to constrain the transport and degradation time for terrestrial OC (terrOC) on the Laptev Sea shelf. The results presented in this manuscript provide important insights about the fate of terrOC in a rapidly changing system (Arctic Shelf). I have read the revised manuscript and author responses to the initial round of reviewer comments and find that the authors have addressed the reviewer comments satisfactorily.

As stated in my previous review, this manuscript provides a unique and interesting data set that provides new insights about the transport time and fate of terrOC across the Siberian-Arctic shelf. This is important work and will make an excellent contribution to Nature Communications.

Detailed author responses to reviewer comments, and edits to manuscript number NCOMMS-17-20661A "Bounding cross-shelf transport time and degradation in Siberian-Arctic land-ocean carbon transfer" by L. Bröder, T. Tesi, A. Andersson, I. Semiletov, and Ö. Gustafsson

Reviewer #2

SPECIFIC POINTS:

- 1) *"Line 21- I feel like the sentence starting with "Here, we leverage..." is missing the object of what you are specifically leveraging, i.e. the utilization of a method to address a gap in knowledge. Perhaps would read better as "Here, we leverage the utility of compound-specific..."*

We have changed the word "leverage" to "employ" for clarification. Due to the abstract word limit we could not add the full suggested edit.

- 2) *"Line 24- Perhaps would be better as "TerrOC, as determined through isotopic mixing model, was..."*

This is a good clarifying point, alas, as mentioned above, word limitations do not allow for this revision to be included.

- 3) *"Line 25-26- "long-lasting" is a bit vague. Perhaps better as "cross-shelf" and/or "net"?"*

We have changed "long-lasting" to "cross-shelf".

- 4) *"Line 34- do you need to redefine "terrOC" here, as it has only been defined in the Abstract?"*

We have followed this suggestion and added the explanation of the abbreviation. It is now: "In high-latitude regions, increasing soil permafrost thaw³, accelerating coastal and sea floor^{4,5} erosion, and rising fluvial sediment discharge⁶ are expected to amplify the delivery of terrestrial organic carbon (terrOC) to the Arctic Ocean."

- 5) *"Line 41-42 I would say that this also depends on the type of shelf! Might be useful at some point in introduction or conclusions to tie in the idea that this type of work can be applied to other systems to better understand variability in these cross-shelf processes on terrOC "sequestration" Also, see comment on lines 210-216 below in regards to this suggestion."*

This is a good point and one of our main conclusions. We believe this message is conveyed clearly by (lines 224-226 and 228-231 in the revised manuscript): "...the current results document a potentially small, but persistent leakage of terrOC to short-term reservoirs on large continental margins, which stands in contrast to the rapid terrOC burial reported for e.g., tropical mountain rivers⁴⁷ or fjords around the globe⁴⁸. [...] For the Mackenzie River, the

efficient transfer and burial of terrOC has been suggested as a geological CO₂ sink⁹. For the wide Eurasian Arctic shelves, on the other hand, long-lasting sediment transport allows for terrOC degradation, thereby constituting a carbon source to overlying water and atmosphere.”

6) *“Line 45 “attributed to” instead of “due””*

We have revised that sentence accordingly.

7) *“Line 48 comma after the citations “15,16” and “in” instead of “of””*

We have now made the suggested changes.

8) *“Line 52- “the fate of terrOC” instead of “terrOC fate””*

This has been changed accordingly.

9) *“Line 67- Omit “here””*

This word has been moved, see below.

10) *“Line 68- add “here” before “we””*

We have moved the word “here” to the position suggested by the reviewer.

11) *“Line 104- Maybe give an average and standard deviation of the 13C values of the LCFA here?”*

We have now included the average d¹³C value and standard deviation: (-31.2 ± 0.5‰) (line 102 in the revised manuscript).

12) *“Line 106- I would recommend removing “Isotopic ratios of...”, and start the sentence with “Stable hydrogen isotopes...””*

This sentence has now been revised as suggested.

13) *“Line 129- “Additionally” instead of “Also””*

We have made the suggested change.

14) *“Line 131- Add “Therefore” to start the sentence “The observed cross-shelf...””*

This word has been moved to the start of the sentence.

15) *“Line 132- add something along the lines of “as opposed to variations in source or hydrodynamic sorting” to the end of this sentence”*

This sentence now reads as follows: “Therefore, the observed cross-shelf increase in LCFA ages can be attributed to terrOC ageing during lateral transport as opposed to variations in source material or hydrodynamic sorting.” (revised manuscript line 128-130).

16) *“Line 136- Add “net transport time” after “quantitative estimate””*

We have now added “for the net transport time” at the end of this sentence.

17) *“Lines 157-158 and lines 182-184 seemingly contradict each other. Clarify what is meant in each statement to resolve this apparent contradiction.”*

We have revised lines 182-184 to now read as follows:” While these rates are all similar, the relative difference between the different carbon pools are consistent with previous findings.” in order to resolve that apparent contradiction.

18) *“Lines 187-195- These lines address one of the main concerns of reviewer #3, however, I feel the explanation in the text is still a bit confusing to follow. Perhaps can alter the order this is presented from the analytical progression you took to its overall implications - i.e. I understood this section as follows: you included the recalcitrant component to deal with the idea of multiple reactivates across the shelf, perhaps due to OET, mineral protection, etc., as evident by the 3X rate of degradation that might be occurring within the first 1.5kyr that the material is transported, compared to the net transport time of 3.6kyr of terrOC transport across the shelf.”*

We agree this could be explained even better and have revised the text. Following the reviewer’s suggestion, we have now changed the order of the arguments around in order to clarify this paragraph. It now reads: “Furthermore, we decided to include a recalcitrant component (i.e., a fraction with a degradation rate constant of 0 kyr^{-1}) to account for the limited changes observed for transport times $>1.5 \text{ kyr}$. When computing degradation rate constants for the subset of inner-mid shelf data points where the transport time is shorter than 1.5 kyr, the rate constants are on average a factor of three higher than when using the full set of observations across the entire shelf (Supplementary Table 2).” (revised manuscript lines 181-188).

19) *“How exactly do the three sentences from lines 201-216 tie together? I would perhaps address this as how estimates of cross shelf transport times benefit from these types of analyses e.g. “Previous attempts to quantify cross-shelf transport times in Washington margin using volcanic ash from the 1980 Mount St. Helen eruption (ref 46) may underestimate true terrOC cross-shelf transport times comparatively to bulk OC radiocarbon techniques (ref 25). Furthermore, the utilization of compound-specific biomarker analyses presented in this study allows for [pick up on line 216]....” This might also be better incorporated if included in the introduction instead, as motivation for why a better approach is needed for quantitatively estimating cross shelf transport times.”*

We have changed this paragraph to improve its connectivity to read as follows: “An earlier attempt to quantify cross-shelf transport times on the Washington margin by tracing the volcanic ash of the 1980 Mount St. Helen eruption resulted in a transport time of <1 year²¹. This study may have underestimated the true terrOC transport time as this value stands in sharp contrast to another estimate using bulk organic carbon ¹⁴C measurements and assumptions on the proportion of terrOC in the bulk to derive a cross-shelf transport time of approximately 1800 years²². Furthermore, the use of compound-specific radiocarbon dating, as in our study on LCFAs in the mobile fraction (<63 μm) along the 600-km long Laptev Sea transect, circumvents uncertainties caused by e.g., changing proportions of marine organic matter and hydrodynamic sorting, and thus enables a quantitative constraint for terrOC transport across one of the widest ocean margins on Earth.” (revised manuscript lines 204-213).

20) *“Line 228- Add “on large continental margins””*

This sentence has been revised accordingly.

21) *“Line 235- The effect of what? Add “terrOC has...” between “effect” and “on””*

We have also changed this sentence and added “of terrOC” between “effect” and “on”.

22) *“Line 237- start the sentence “Sediment...” with a “Therefore,” Or “Thus,” Or “In conclusion,””*

We have revised the sentence to now start with “In conclusion, ...”.

23) *“Figure 3. I would just reiterate in this figure caption (especially C-F) that the biomarker data was corrected for what was 'present' in the fine fraction, which is outlined in Figure S5.”*

This sentence has been changed accordingly.